# Developmental activities of the complement pathway in migrating neurons

Anna Gorelik[1], Tamar Sapir[1], Rebecca Haffner-Krausz[2], Tsviya Olender[1], Trent M. Woodruff[3] & Orly Reiner[1]

In recent years the notion that malfunctioning of the immune system may result in developmental brain diseases has emerged. However, the role of immune molecules in the developing brain has not been well explored. The complement pathway converges to cleave C3. Here we show that key proteins in the lectin arm of this pathway, MASP1, MASP2 and C3, are expressed in the developing cortex and that neuronal migration is impaired in knockout and knockdown mice. Molecular mimics of C3 cleavage products rescue the migration defects that have been seen following knockdown of *C3* or *Masp2*. Pharmacological activation of the downstream receptors rescue *Masp2* and *C3* knockdown as well as *C3* knockout. Therefore, we propose that the complement pathway is functionally important in migrating neurons of the developing cortex.

[1] Department of Molecular Genetics, Weizmann Institute of Science, 234 Herzl St, Rehovot 7610001, Israel. [2] Department of Veterinary Resources, Weizmann Institute of Science, 234 Herzl St, Rehovot 7610001, Israel. [3] School of Biomedical Sciences, University of Queensland, BNE, Brisbane, Queensland 4072, Australia. Correspondence and requests for materials should be addressed to O.R. (email: orly.reiner@weizmann.ac.il).

The notion that malfunctioning of the immune system may contribute to autism spectrum disorder (ASD) has been suggested[1,2]. Immune signalling has been shown to participate in postnatal neural development and synaptic plasticity[3]. The innate immune complement pathway is composed of a cascade of proteases. Upon stimulation, an amplifying cascade of cleavages begins resulting in cell surface bound and receptor activating molecules, ultimately leading to cell destruction as a defence response to pathogens. In the postnatal brain, activity of the classical complement pathway has been implicated in developmental pruning of synapse refinement of the mouse visual system[4–7]. In addition, it has been shown that the complement fragment C3a and its receptor are important for collective cell migration of neural crest cells[8]; however, the role of the complement system in the embryonic brain remains poorly understood. Mutations in members of the lectin arm of the complement pathway have been previously implicated in 3MC syndrome[9,10], in which intellectual impairment is part of the complex syndrome. Neuronal migration is impaired in cases of intellectual disabilities, autism and schizophrenia[11,12], and therefore, we postulated that the complement cascade might be involved in regulation of migrating neurons in the developing brain. Here we show that key proteins in the pathway, Complement C3, MASP1 and MASP2, are expressed in the developing mouse brain and that neuronal migration is impaired when their levels are reduced by either knockout or knockdown. Molecular mimics of C3 cleavage products rescued knockdown of C3 or Masp2. Furthermore, pharmacological activation of the downstream receptors C3aR and C5aR rescued the migration defect caused by Masp2 and C3 knockdown as well as C3 knockout. Collectively, our data suggest that complement activity is required for neuronal migration progression, and thus a beneficial intervention is possible.

## Results

**Complement components in the brain affect neuronal migration.** Complement C3 is a central molecule in the complement pathway and it is cleaved following the activation of either the classical-, or the lectin-, or the alternative arms of the complement pathway (the classical- and lectin-arms are depicted in Fig. 1a). In the lectin-arm, the two key proteases that are activated are MASP1 and MASP2. Immunostaining demonstrated widespread immunoreactivity of C3, MASP1 and MASP2 in embryonic brain sections (E16, Fig. 1b–d). Relatively low level of expression was noted in the ventricular zone (VZ), and the subventricular zone, where Tbr2 positive cells reside (Fig. 1e). Expression of C3, MASP1 and MASP2 was noted in areas where postmitotic neurons are present (Tuj1 positive, Fig. 1f), and higher levels of expression were detected in the cortical plate (CP), where Tbr1 immunostaining marks the deep layers (Fig. 1g). The validity of the C3, Masp1 and Masp2 antibodies was confirmed by showing loss of immunoreactivity in brain sections of the respective knockout (KO) mice (Supplementary Fig. 1a–h). To complement the protein expression data, we examined the expression of the mRNA of the abovementioned molecules in cortices of developing embryos. In addition, we tested for mRNA expression of two important receptor proteins in this pathway, C3aR and C5aR. C3aR is the receptor for C3a, the cleavage product of C3, while C5aR binds C5a. The mRNA expression of the complement genes C3, Masp1, Masp2, C3aR and C5aR was dynamic and varied at different embryonic ages (Supplementary Fig. 1i–m). To better visualize the cellular localization of the complement proteins, E14 mouse embryos were in utero electroporated with Lifeact-GFP that marks the cell periphery, and the brain sections were immunostained at E16.

Neuronal progenitors that are electroporated in the VZ will differentiate and follow the normal migratory path to superficial layers of the CP at E18 (Fig. 1h). At E16 the majority of migrating neurons electroporated on E14 undergoes multipolar-to bipolar transition in the intermediate zone (IZ). High resolution and Z-stack images of neurons taken from the IZ (relative position marked by * in Fig. 1b–d) demonstrate that most of the complement protein deposits are extracellular (Fig. 1i–k).

We next questioned whether the pathway is active in the developing brain. To this end, we tested and showed proteolytic processing of Complement protein C3 using an ELISA kit specific for the detection of cleaved C3 (C3a); we detected $2.89 \pm 0.32$ ng of cleaved C3 per mg of total protein with no C3a expression in C3 knockout cortices. In addition, C3b was detected in developing brain lysates by western blot analysis using antibodies specific for activated fragments of C3 (Supplementary Fig. 1n). We then tested the hypothesis that C3 can regulate radial neuronal migration in the developing brain. The position of C3 knockout labelled neurons across the width of the cortex differed significantly from that found in the wild type (Fig. 1l,m). In the C3 knockout brains more GFP labelled neurons were found in deeper bins (closer to the VZ) and less were found in the superficial layers of the CP suggesting neuronal migration impairment. Following this, the possible role of members of the lectin pathway in regulation of neuronal migration was studied. The lectin pathway is activated via the binding of the recognition molecules mannose-binding lectin (MBL) and of ficolins, to different ligands. The binding activates the associated proteases mannan-binding lectin serine peptidases 1 and 2 (MASP1 and MASP2) (reviewed in refs 13,14). Neuronal migration was analysed in Masp1 knockout embryos generated by CRISPR/Cas9 gene editing (Supplementary Fig. 1c,d,g). Neurons born at E14 were labelled with IdU, and their relative position in the brain was analysed at E18 (Fig. 1n,o). While control litter-mates exhibited a normal pattern of neuronal migration with most of the E14 labelled neurons reaching the CP by E18, neurons in $Masp1 -/-$ embryos had a wider pattern of distribution (differences are significant in three out of five bins, Fig. 1o). In a similar manner, the role of MASP2 in migrating neurons was examined, using $Masp2 -/-$ embryos generated by CRISPR/Cas9 gene editing (Supplementary Fig. 1e,f,h). Neuronal migration impairment was observed in the Masp2 knockout embryos (Fig. 1p,q). Comparison of the position of neurons labelled by IdU at E14 and analysed at E18 revealed significant differences between Masp2 knockout and their wild-type litter-mates (three out of five arbitrary bins across the width of the CP, Fig. 1q). We next tested for possible cortical lamination phenotypes in the abovementioned knockout mice. We noted that the width of C3 knockout cortices was reduced in comparison to control ($76.4 \pm 1.4\%$, versus $100 \pm 3.5\%$, $n = 8$, Student's t-test, $P = 0.0004$). The relative width of the deep-layers (Tbr1+) and the superficial layers (Cux1+) increased by $4.8 \pm 1.3\%$ ($n = 8$, Student's t-test, $P = 0.03$) and $16.8 \pm 3.4\%$ ($n = 8$, Student's t-test, $P = 0.003$), respectively (Fig. 1r–t). In case of Masp1 and Masp2 knockout, a reduction in the relative domain occupied by Cux1+ cells was noted ($14.6 \pm 2.3\%$, $n = 5$, Student's t-test, $P = 0.0017$ and $10.3 \pm 2.8\%$, $n = 5$, Student's t-test, $P = 0.022$ respectively, Fig. 1u–z). No change in the Tbr1-positive domain was observed. The width of Masp1 knockout cortices was similar to that of wild-type, whereas Masp2 knockout cortices exhibited an 8% reduction in width as compared to wild-type ($92 \pm 2.6\%$ versus $100 \pm 1.9\%$ respectively, $n = 5$, Student's t-test, $P = 0.04$).

We next moved to a knockdown system that allows an acute downregulation of gene products in a small subpopulation of neurons, thus avoiding gene-redundancy effects and serving as

a module for screening for possible rescue molecules. *C3* knockdown in the developing cortex resulted in a neuronal migration deficiency (Fig. 2a,b,e). The efficiency of C3 shRNA was confirmed by qPCR and was found to reduce the mRNA levels dramatically to $14.5 \pm 10.6\%$ ($n = 6$, Student's *t*-test,

$P = 0.009$) of the control expression levels. C3 protein levels were consequently reduced as evident by immunostaining, and were corrected when a C3 cDNA construct resistant to the shRNA (C3$^{res}$) was co-introduced (Supplementary Fig. 2a–d). *In utero* electroporation of *Masp1* shRNA significantly impaired

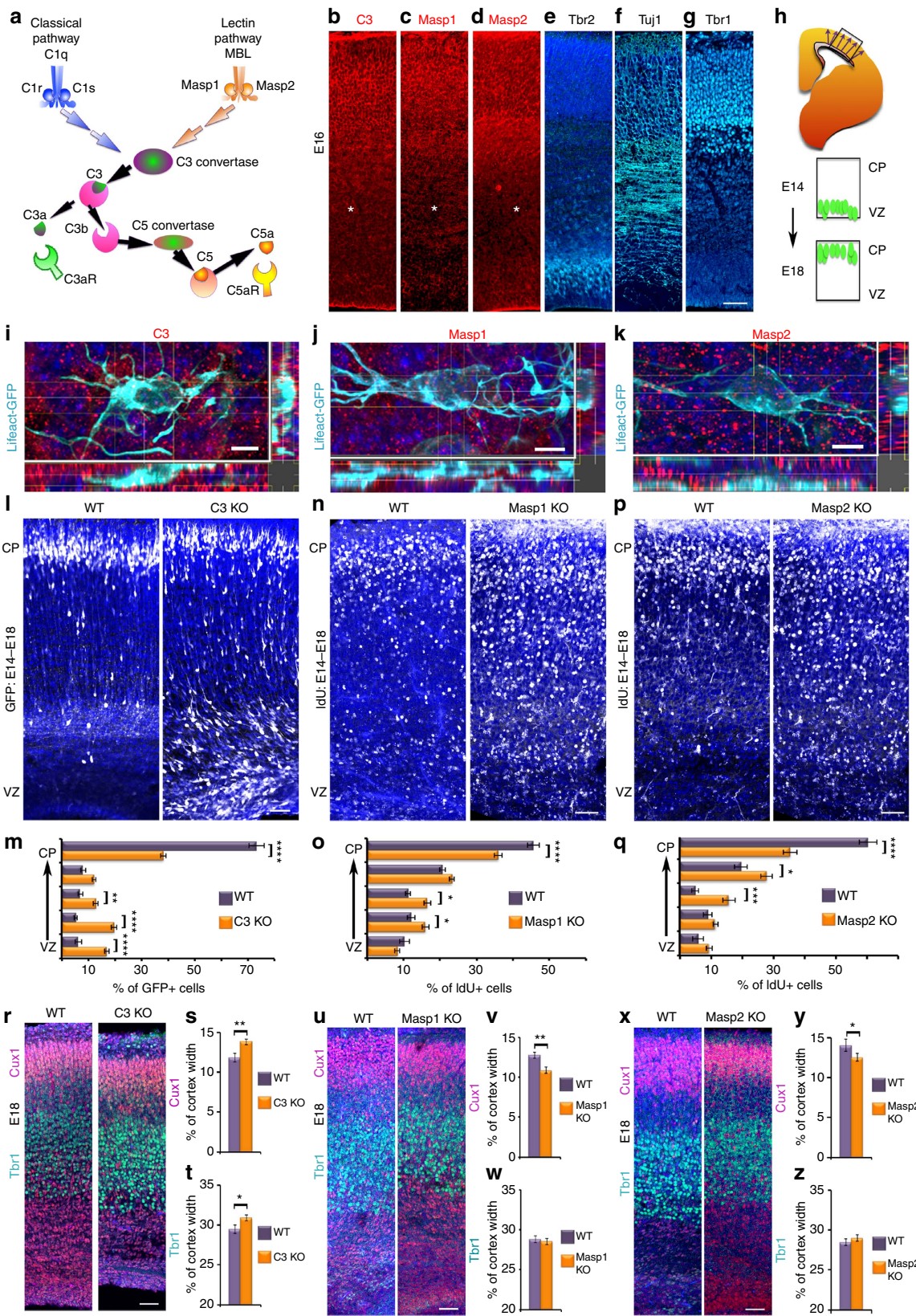

neuronal migration (Fig. 2c, Supplementary Fig. 4a–e), recapitulating our finding in the knockout mice. *Masp1* shRNA reduced *Masp1* mRNA levels by 38.5 ± 3.5% in comparison to control shRNA (real-time qPCR, $n = 9$, Student's *t*-test, $P = 4.8E^{-08}$.) *Masp1* shRNA treatment also resulted in decreased MASP1 immunoreactivity (Supplementary Fig. 2e,f,h). Addition of *Masp1*$^{res}$ partially corrected the observed protein levels (Supplementary Fig. 2e–h). Neuronal migration impairment was also observed in the embryonic brains treated with *Masp2* shRNA (Fig. 2a,d,e). The position of the labelled neurons significantly differed in all five bins, spanning the width of the CP. *Masp2* shRNA levels were reduced by 52.4 ± 7.4% in comparison to control ($n = 9$, Student's *t*-test, $P = 7.1E^{-05}$). The changes in MASP2 protein levels following shRNA or rescue treatments were confirmed by immunostaining (Supplementary Fig. 2i–l). The shRNA treatments (E14–E18) of the three complement protein genes resulted in neurons that are stalled in the IZ. The morphology of the stalled cells combined with immunostaining for Golgi was analysed. In polarized migrating neurons, the Golgi apparatus clusters and usually only one cluster per cell is detected. When either *C3, Masp1* or *Masp2* shRNA were introduced, a dispersion of the Golgi apparatus was evident on E18 with more than two clusters per cell, corresponding to cells which exhibited multipolar morphology (Supplementary Fig. 2m–q). The multipolar morphology is a transient state that is normally followed by adoption of bipolar morphology. Whereas control E14-labelled cells undergo multipolar-to-bipolar transition on E16, this process was impaired following knockdown of either *C3*, or *Masp1* or *Masp2*. We examined whether *Masp2* knockdown affected apoptosis, but the number of apoptotic cells on E18 did not differ from control treatment (Supplementary Fig. 3). Next, the postnatal effects of the abovementioned embryonic interventions were studied. Analysis of the position of Cux1-positive cells on P8 revealed that knockdown of either *C3*, or *Masp1* or *Masp2* resulted in a more superficial position than the control cells. In case of either *Masp1* or *Masp2* shRNA a significant proportion of the cells did not reach superficial layers at all. Some of the *Masp2* shRNA cells were positioned ectopically in the white matter. In case of *Masp1* shRNA treatment, cells abnormally positioned in the IZ/CP border were noted (Fig. 2f–n). To determine the fate of the cells in ectopic positions, sections were immunostained with antibodies for the upper layer marker Cux1 and the deep layer marker Tbr1. The majority of cells in control shRNA treated brain were Cux1 positive (94.8 ± 0.8%, Fig. 2f,j, Supplementary Fig. 2v). The majority of *C3, Masp1* and *Masp2* shRNA treated cells were Cux1 positive regardless of their position (91.4 ± 0.8%, 86.9 ± 1%, $P = 0.01$, 78.6 ± 1.6%, $P = 0.01$ respectively, Fig. 2g–i, Supplementary

Fig. 2v). The reduction of the total number of Cux1-positive cells following *Masp1* or *Masp2* knockdown was not compensated by additional cells positive for the deep layer neuronal marker Tbr1 (Fig. 2j–m, Supplementary Fig. 2w) nor by the number of cells positive for the glial marker GFAP (Supplementary Fig. 2r–u,x).

We next examined whether the observed phenotypes could be solely explained by changes in the proliferative behaviour of the progenitors (Supplementary Fig. 5). We tested neuronal stem cell proliferation by a short IdU pulse, which is incorporated in S phase (30 min), and phospho-Histone H3 (pHis +), preferentially labelling cells in M phase, and immunostained embryonic brain sections using the respective antibodies. No differences between *C3* knockout and littermate control embryos were observed in the amount of pHis + cells in the VZ (Supplementary Fig. 5e–g). However, in the subventricular zone of *C3* knockout brains, an increase of 24.8 ± 2% in pHis + cells was observed in comparison to the control littermates (Supplementary Fig. 5e–g). These results suggest possible changes in cell-cycle parameters of the intermediate progenitors. The same assays were conducted in embryos treated with *Masp1* or *Masp2* shRNA constructs compared with control shRNA at E13 and analysed on E14. No changes were noted in case of *Masp1* shRNA treated brains in the proportion of IdU + (Supplementary Fig. 5a,b,d). However, *Masp2* shRNA treatment resulted in a significant increase of IdU + cells (increase of 47.9 ± 7%, Supplementary Fig. 5a,c,d). The observed changes in the proliferation of the progenitors may explain in part the reduction in the width of *C3* and *Masp2* knockout cortices mentioned above; however, since the migratory effect seen across treatments is not consistent with the proliferation abnormalities, it is unlikely that the observed migration phenotype can be exclusively explained by the changes in cell cycle parameters.

The initiation of the classical arm of the complement pathway requires the binding of C1q to many of its interacting partners, which then leads to a conformational change in the C1 complex and to the exposure of the serine proteases C1r and C1s to cleave C2 and C4 (reviewed in ref. 13). *In situ* hybridization data (GenePaint.org) show the expression of C1q, C1s and C1r in the developing brain. To test the role of classical complement cascade in corticogenesis, we examined whether C1qa-deficient mice[15] (a gift from Marina Botto, Imperial College London) exhibit a neuronal migration phenotype. At E18, the position of the GFP-positive neurons born at E14 did not differ between the wild-type and the knockout littermates (Supplementary Fig. 6a–c). Most of the GFP-positive neurons resided in the expected CP layers in brain sections from embryos of both genotypes. Knockdown of an additional component of the

**Figure 1 | Complement components are expressed in the developing brain and affect neuronal migration.** (**a**) Schematic presentation of the complement pathway. (**b–g**) Immunostaining of cortices (E16) with anti-C3, anti-MASP1 and anti-MASP2 antibodies (**b–d**, respectively) demonstrate wide expression of these proteins. Immunostaining with anti-TBR2, anti-TUJ1 and anti-TBR1 antibodies (**e–g**, respectively) show the borders of the ventricular zone, intermediate zone and cortical plate, respectively. (**h**) Scheme showing the route of radial migration in the cerebral cortex. Neurons born at E14 in the VZ (green cells) migrate and on E18 are located at superficial extremity of the cortex. (**i–k**) Extracellular localization of the tested complement proteins. Brains were *in utero* electroporated (E14–E16) with Lifeact-GFP to mark the cell periphery. Sections were immunostained with anti-C3, anti-MASP1 and anti-MASP2 antibodies (**i–k**, respectively). High-magnification 3D reconstructions of individual neurons from the IZ were performed using confocal microscopy and IMARIS software. The asterisks in **b–d** demonstrate the approximate position of the chosen neurons. (**l–q**) *C3, Masp1* and *Masp2* KO affect neuronal migration. (**l,m**) WT or *C3* KO embryos were *in utero* electroporated with GFP (E14–E18). The GFP positive neurons in the *C3* KO ($n = 5$) resided in deeper layers than in the WT ($n = 3$).The position of the GFP + cells was quantified in five bins (from the VZ to the CP) (**m**). (**n–q**) Birth-dating by IdU at E14 and analysis at E18 show that in WT ($n = 3$) sections (**n,p**), labelled cells are located in more superficial layers than in *Masp1* KO ($n = 3$) sections (**n**) or in the *Masp2* KO ($n = 3$) sections (**p**). The distribution of IdU-positive neurons (**o,q**). *C3* KO brain ($n = 8$) sections (**r**), *Masp1* KO brain ($n = 5$) sections (**u**) and *Masp2* KO brain ($n = 5$) sections (**x**) and WT brain ($n = 8$, $n = 5$, $n = 5$ respectively) sections immunostained with anti-CUX1 and anti-TBR1 antibodies. The relative proportion of the CUX1-positive and TBR1-positive domains relative to the total cortical width are displayed (Student *t*-test, **s,v,y** and **t,w,z**, respectively). (**i–k**), scale bars are 5 μm, (**b–g,l,n,p,r,u,x**, scale bars are 50 μm). *$P < 0.05$; **$P < 0.01$; ***$P < 0.001$, ****$P < 0.0001$, histograms present means ± s.e.m.

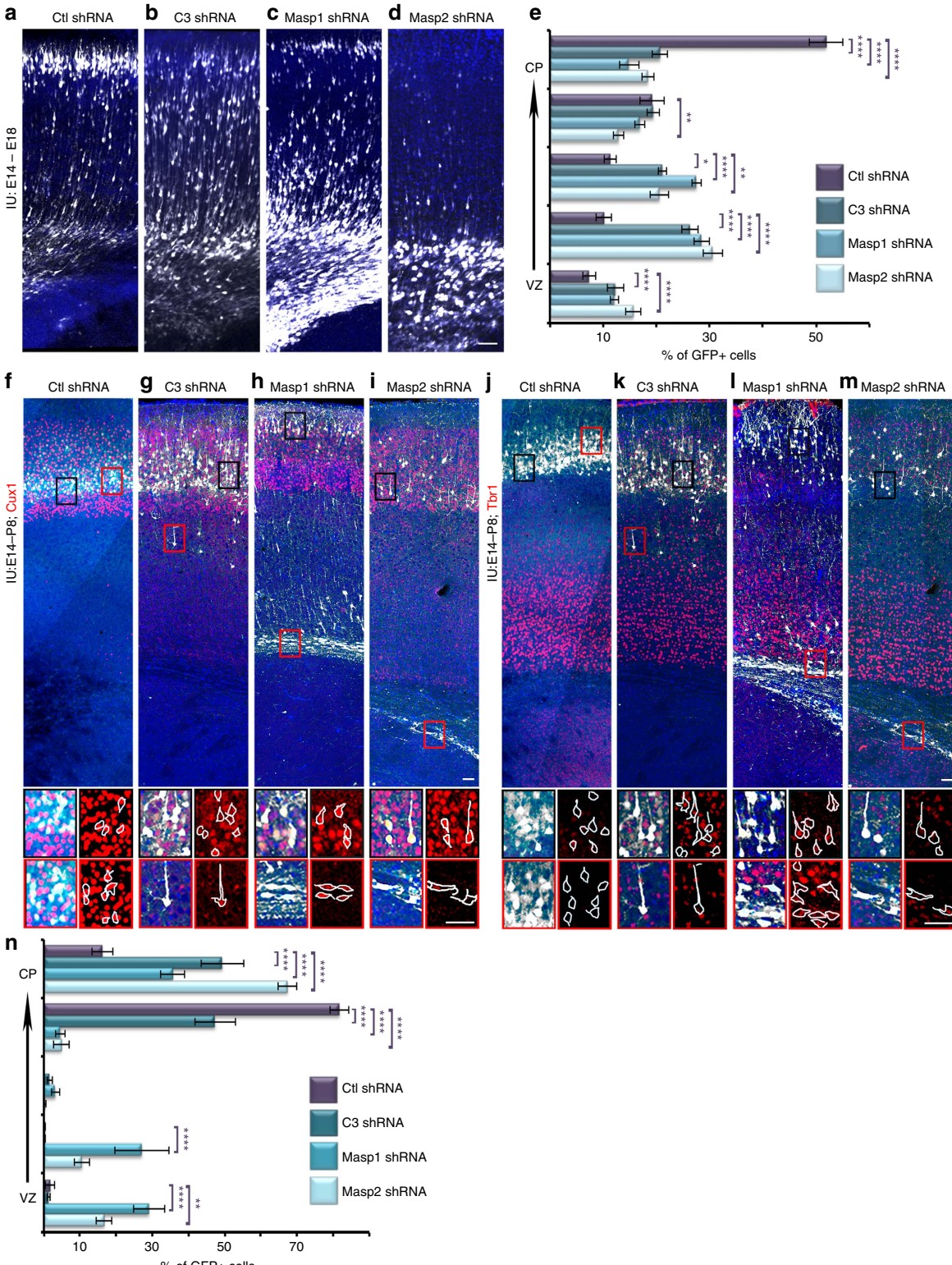

**Figure 2 | C3, Masp1 and Masp2 knockdown affect neuronal migration.** (**a**–**e**) Brains were electroporated *in utero* (E14–E18) with control shRNA (**a**, $n = 4$), *C3* shRNA (**b**, $n = 5$), *Masp1* shRNA (**c**, $n = 5$) or *Masp2* shRNA (**d**, $n = 4$). The scale bars are 50 μm. (**e**) The position of GFP + neurons across the width of the cortex was analysed and is shown in five bins (from the VZ to the CP). All shRNA treatments were compared to control shRNA. (**f**–**n**) Postnatal positioning and the identity of *C3*, *Masp1* and *Masp2* knockdown cells. Brains electroporated *in utero* on E14 with control shRNA (**f,j**), *C3* shRNA (**g,k**), *Masp1* shRNA (**h,l**) or *Masp2* shRNA (**i,m**) were immunostained at postnatal day 8 (P8) with anti-CUX1 or anti-TBR1 antibodies (**f-i,j-m**, respectively). Black and red boxes show the position of the enlargements underneath each slice. The immunostainings for the enlarged areas are shown together with GFP or with the outlines of GFP-positive cells. (**n**) The position of GFP + neurons across the width of the P8 cortex was analysed and is shown in five bins (from the VZ to the CP). All shRNA treatments were compared to control shRNA. The scale bars are 50 μm. *$P < 0.05$; **$P < 0.01$; ***$P < 0.001$; ****$P < 0.0001$, histograms present means ± s.e.m.

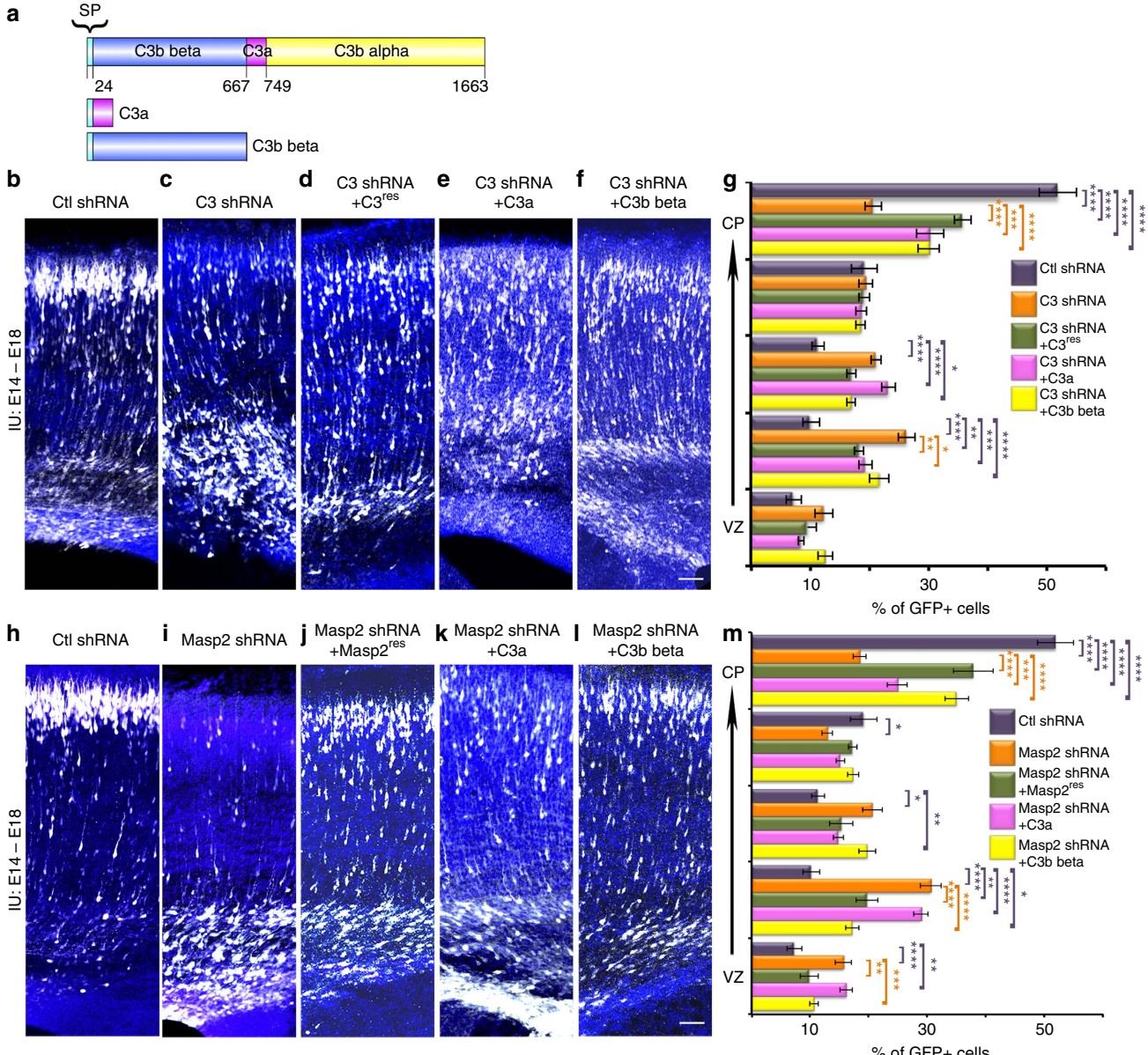

**Figure 3 | C3 cleavage products can rescue upstream migration deficits.** (**a**) Schematic presentation of C3 cleavage (top) and the two peptides used in this study, C3a with an attached signal peptide (SP) and C3b beta. (**b–g**) C3 knockdown impairs neuronal migration and can be rescued in part by addition of either C3res, or C3a, or C3b beta. Brains were electroporated *in utero* (E14–E18) with control shRNA (**b**, n = 4), C3 shRNA alone (**c**, n = 5), or in combination with C3 resistant to the shRNA (**d**, n = 3), C3a (**e**, n = 3) or C3b beta (**f**, n = 4). The distribution of neurons along the cortex is shown in five bins (from the VZ to the CP) for all the treatments (**g**). All the treatments were compared to control shRNA (the statistical significance is shown in violet) and to C3 shRNA (the statistical significance is shown in orange). (**h–m**) Masp2 knockdown impairs neuronal migration via the complement pathway. Brains were electroporated *in utero* (E14–E18) with control shRNA (**h**, n = 4), Masp2 shRNA alone (**i**, n = 4), or in combination with Masp2 resistant to the shRNA (**j**, n = 3), C3a (**k**, n = 4) or C3b beta (**l**, n = 4). Quantification of the distribution of neurons across the cortex is shown for all the treatments (**m**). All the treatments were compared to control shRNA (the statistical significance is shown in violet) and to C3 shRNA (the statistical significance is shown in orange). The scale bars are 50 μm. *P < 0.05; **P < 0.01; ***P < 0.001, ****P < 0.0001, histograms present means ± s.e.m.

classical pathway, *C1s*, resulted in mild though significant neuronal migration inhibition (Supplementary Fig. 6d–f). The reduction of *C1s* mRNA by shRNA expression was verified by qPCR, displaying 47.4 ± 8% reduction in the levels of *C1s* as compared to control shRNA (n = 5, Student's t-test, P = 0.0067).

Whereas knockout mice for the C1qa (a component of the classical pathway) show normal neuronal migration, knockout and knockdown for components of the lectin pathway (*Masp1* and *Masp2*), and complement *C3* exhibit neuronal migration impairment.

**C3 cleavage products rescue migration phenotypes.** Activation of the complement pathway results in enhanced cleavage of C3 to C3a and C3b (reviewed in ref. 16) (Fig. 1a). Therefore, we hypothesized that if C3 proteolysis is important for neuronal migration, the expression of peptides, which mimic the cleavage products may mend neuronal migration deficits caused by the reduction of the level of upstream components of the lectin pathway. *In vivo*, C3b beta and alpha are attached by disulphide bonds, which we could not imitate in mimicry products; thus, to ensure their secretion, a signal peptide was attached to the

N-terminal part of C3a, while C3b beta naturally has the signal peptide (Fig. 3a)[17]. As previously mentioned C3 knockdown resulted in neuronal migration deficiency (Fig. 3b,c,g). Specificity was indicated by successful partial rescue following introduction of an expression construct of C3 resistant to the shRNA (Fig. 3d,g). The C3 mimicry products were then expressed in utero together with C3 shRNA. Addition of the C3a or the C3b beta fragments led to partial prevention of the C3 shRNA

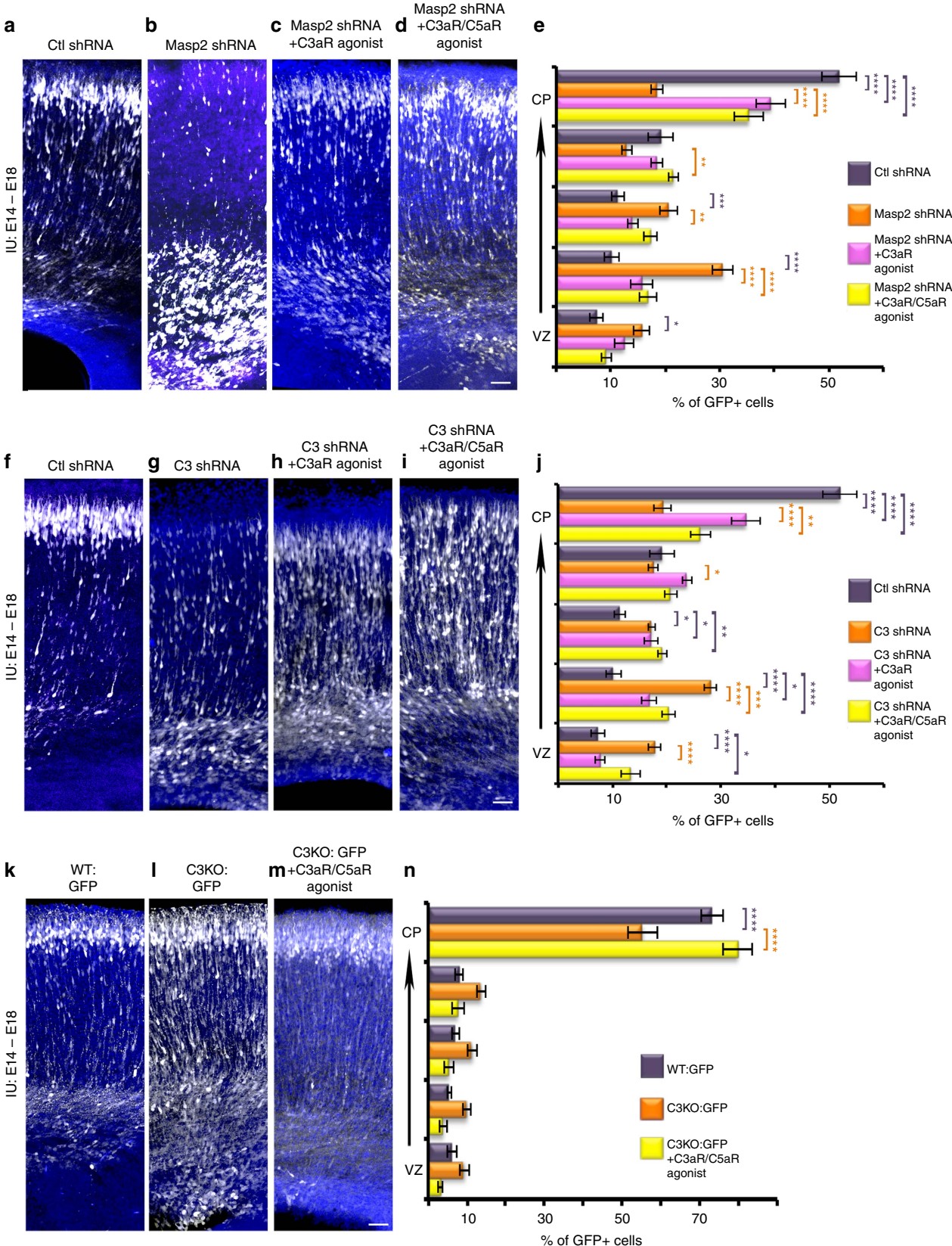

migration phenotype (Fig. 3e–g). The finding that the molecular mimics of C3 cleavage products improved C3-knockdown-induced migration deficits suggests that the activity of the complement pathway, and not only the C3 protein, is required for proper positioning of excitatory neurons in the developing brain. We then tested whether these mimicry products are capable of alleviating neuronal migration impairment when a more upstream component, *Masp2*, is knocked down. *Masp2* was chosen for rescue experiments because its functioning is more restricted to the activation of the lectin pathway in comparison with *Masp1*. Masp1 has been shown to activate the alternative pathway[18] and even to directly cleave C3 (ref. 19). As mentioned above, neuronal migration impairment was observed in the embryonic brains treated with *Masp2* shRNA (Fig. 3h,i,m). The position of the labelled neurons significantly differed in all five arbitrary bins spanning the cortex. This phenotype was markedly improved following the addition of *Masp2* resistant to the shRNA (*Masp2*[res]) (Fig. 3j,m). Next, C3a or C3b beta peptides were expressed *in utero* together with *Masp2* shRNA (Fig. 3k–m). The expression of C3a to some extent, and of C3b beta to a better extent, significantly improved the displacement of the GFP-positive neurons.

**Prenatal treatment with C3a/5a receptor agonists**. We further hypothesized that receptors for C3a and/or C5a may be the downstream signalling components mediating the complement activity within the developing brain (schematic presentation of the pathway in Fig. 1a). The C3a and the C5a receptors are expressed in the developing brain as evident by RNA *in situ* hybridization data of E14 brain from GenePaint.org and real-time RT–PCR from E13–E18 cortices (Supplementary Fig. 1l,m). The relative levels of C3aR and C5aR in the E16 cortex are $14.7 \pm 2.4\%$ and $2.9 \pm 0.4\%$, respectively, in comparison to the levels of C3aR and C5aR in adult mouse liver. To test our hypothesis, we introduced either a selective C3aR agonist[20], or a dual C3aR/C5aR agonist[21,22] together with *Masp2* knockdown (Fig. 4a–e). *Masp2* knockdown is expected to result in less activation of the C3a and C5a receptors due to the reduction in the level of C3 cleavage products. Addition of either the single C3aR agonist or the dual complement peptide receptor agonist significantly rescued neuronal migration impairment (Fig. 4a–e). The dosage of the agonist treatment was determined using dose change of agonist. Half dose of the agonist partially rescued the migration phenotype combined with *Masp2* shRNA treatment. However, half-dosage was significantly less effective than the full dose treatment (Supplementary Fig. 7). Our hypothesis predicted that the activation of the C5a and/or C3a receptors would rescue the phenotype caused by *C3* shRNA treatment. Indeed, to a large degree the dual agonist, and to a lesser degree the C3aR agonist, significantly improved the position of the electroporated neurons (Fig. 4f, control, Fig. 4g, C3 shRNA versus Fig. 4h and i respectively, quantified in 4j). Moreover, we were able to confirm that the activation of the C3a and the C5a receptors can also

alleviate the migration deficit observed in the $C3 - / -$ embryos, while no migration impairment was observed in any of the agonist-treated brains (compare Fig. 4k, wild-type to Fig. 4l, C3 KO versus Fig. 4m, C3 KO with agonists, quantified in Fig. 4n).

Collectively, these results suggest that neuronal migration is orchestrated mainly through activation of the lectin arm of the complement pathway. C3 cleavage occurs in the developing brain and activation of the complement C3a and C5a receptors is required for proper neuronal migration progression.

## Discussion

Our studies show that the active lectin/MASP arm of the complement pathway, leading to cleavage of C3, and activating the complement peptide receptors for C3a and C5a, is required for proper migration of neurons in the developing brain. Intervention in several components of this pathway, *Masp1*, *Masp2* or *C3* substantially impaired neuronal migration. The developmental defects following knockdown treatments had postnatal manifestations and long-lasting effects on the cortical organization. Following knockdown treatments of *Masp1*, *Masp2* or *C3*, labelled neurons occupied a more superficial layer than those observed in the control treatment. In addition, ectopic neurons were found in deep cortical layers or in even in the white matter following *Masp1 or Masp2* knockdown respectively. In case of *Masp1* knockdown the majority of the ectopic cells were Cux1 positive, whereas in case of *Masp2* knockdown only a portion of the ectopic cells exhibited Cux1 immunoreactivity. In both treatments, the ectopic cells were Tbr1 and GFAP negative. Therefore, although the treatments resulted in improper positioning of the labelled II–IV neurons born at the time of electroporation or IdU incorporation, the phenotype varied according to the specific gene being manipulated. This may be explained in part by the multi-arm nature of the complement pathway, the complex activities of these proteins and their interactions with additional pathways. It may be possible to speculate that *C3* related phenotypes can be partially rescued by C5 convertase activity of thrombin as has been previously demonstrated in *C3* knockout mice[23]. Such a compensatory mechanism is far less likely in the case of Masp2 manipulations, since *Masp2* knockdown or knockout is expected to result in less active thrombin[24]. Finally, Masp1 possess an additional active role in the alternative arm of the complement pathway[25], which may be additionally required for proper brain development. To a great extent, addition of molecular mimics of C3 cleavage products prevented neuronal migration inhibition following knockdown of *Masp2* or *C3*. This suggests that functional activation of the pathway, resulting in C3 cleavage and production of C3 and C5 bioactive mediators, may be an important step in shaping the journey of neurons on their way to the CP. Furthermore, pharmacological activation of the C3a and the C5a receptors prevented neuronal migration deficits in case of *C3* or *Masp2* knockdown and exhibited complete restoration of

**Figure 4 | Neuronal migration impairment by either knockdown of *Masp2* or *C3*, or knockout of *C3* is ameliorated by activation of the C5a and/or the C3a receptors.** (**a–e**) Treatment by *Masp2* shRNA (**b**, $n = 4$) electroporated *in utero* (E14–E18) is partially rescued by addition of C3aR agonist (**c**, $n = 5$) or the dual receptor agonist (**d**, $n = 3$). Quantification of the distribution of neurons across the cortex is shown for all the treatments (**e**). All the treatments were compared to control shRNA (**a**, $n = 4$, the statistical significance is shown in violet) or *Masp2* shRNA (the statistical significance is shown in orange). (**f–j**) Knockdown of *C3* (**g**, $n = 4$) electroporated *in utero* (E14–E18) is partially rescued by either addition of C3aR agonist (**h**, $n = 4$) or the dual receptor agonist (**i**, $n = 4$). Quantification of the distribution of neurons across the cortex is shown for all the treatments (**j**). All the treatments were compared to control shRNA (**f**, $n = 4$, the statistical significance is shown in violet) or *C3* shRNA (the statistical significance is shown in orange). (**k–n**) Knockout of *C3* is rescued by addition of the dual agonist. *C3* KO embryos were electroporated *in utero* (E14–E18) with a GFP expression construct (**l**, $n = 6$), or with a GFP expression construct together with the C3a/C5a receptors agonist (**m**, $n = 4$). None of the agonist treated brains showed any neuronal migration impairment. Comparison to the WT (**k**, $n = 3$) is shown in violet. Comparison to *C3* KO is shown in orange, histograms present means $\pm$ s.e.m.

**Table 1 | A list of *de novo* mutations identified in ASD patients.**

| Gene | Location | Substitution | Effect | Amino acid change | Transcript | ExAC_freq | SIFT | PolyPhen |
|---|---|---|---|---|---|---|---|---|
| CR1* | chr1:207793410 | C->T | missense | R1968C | NM_000573 | 1.31E-04 | T | P |
| CR1L* | chr1:207874932 | G->A | missense | G433R | NM_175710 | NA | T | D |
| CD59* | chr11:33731767 | C->T | missense | E98K | NM_001127223 | 1.63E-05 | T | B |
| C1R* | chr12:7188474 | C->G | missense | D494H | NM_001733 | NA | D | D |
| ITGAM† | chr16:31341200 | G->A | missense | D984N | NM_000632 | NA | T | B |
| COLEC12* | chr18:321720 | C->A | missense | W717C | NM_130386 | NA | D | D |
| COLEC12* | chr18:348118 | C->T | missense | R76H | NM_130386 | 3.25E-05 | T | B |
| C3† | chr19:6709711 | T->G | missense | K610T | NM_000064 | NA | D | D |
| C3‡ | chr19: 6678390 | G->C | missense | D1569E | NM_000064 | NA | D | D |
| CPAMD8* | chr19:17104231 | G->T | missense | L468M | NM_015692 | NA | D | D |
| ITGB2† | chr21:46309390 | C->T | missense | G560R | NM_000211 | 2.49E-05 | T | D |
| MASP1* | chr3:186970968 | A->T | missense | Y294N | NM_001031849 | NA | D | D |
| CFI† | chr4:110667606 | G->T | missense | H401N | NM_000204 | 7.32E-05 | T | B |
| FGA† | chr4:155507718 | T->C | missense | N288S | NM_000508 | NA | T | B |
| FCN2* | chr9:137779166 | A->T | missense | T245S | NM_015837 | NA | T | B |
| CD109† | chr6:74440139 | TCTAATAGT->Del | codon_del | 117_119del | NM_001159587 | NA | NA | NA |

The list was downloaded from SFARI base and is an integration of *de novo* mutations found in Iossifov et al.[39], Krumm et al.[38] and Neale BM et al.[40]. ExAC_freq, the frequency of the mutation in the Exome Aggregation Consortium (ExAC), Cambridge, MA, USA (URL: http://exac.broadinstitute.org). SIFT, PolyPhen, Functional prediction of missense mutation using SIFT (PMID:26633127) and PolyPhen2 (PMID:20354512). Possible predictions are: T, tolerant; B, benign; P, possibly damaging; D, damaging; NA, not determined. COLEC12 was identified as a gene with recurrent *de novo* mutations in the study of (PMID:25363768), C3 and FGA have additional specific information for involvement in ASD. COLEC12 was identified as a gene with recurrent *de novo* mutations in the study of (PMID:25363768), C3- was found to be hypomethylated and over expressed in individuals with ASD (PMID: 25180572), FGA- was found to be in association with ASD risk (PMID:24192574).
*Iossifov et al.[39].
†Krumm et al.[38].
‡Neale BM et al.[40].

the position of neurons in *C3* knockout brains, suggesting that a single intervention may be sufficient to ameliorate migration phenotypes. Activation of the C3a or the C5a receptors is likely to activate small GTPases, which will in turn affect cytoskeletal dynamics required for proper neuronal migration. As key regulators of actin and microtubule cytoskeletons, cell polarity and adhesion, the Rho GTPases play critical roles in CNS neuronal migration (reviewed in refs 26,27). The complement fragment C3a and its receptor have been previously demonstrated to act during collective cell migration of neural crest cells[8]. The mode of collective migration is rather different from radial neuronal migration studied here, although our studies imply that at least one of the chemo-attractants may be shared in both processes. In addition, perturbations in C5aR signalling during rodent brain development have been reported to result in select defects[28,29], suggesting widespread roles for complement fragments C3a and C5a in neuronal development.

Knockdown or knockout of specific elements of the classical pathway (*C1qa* and *C1s*) exhibited a mild effect on the progression of neuronal migration. The classical pathway is initiated by the creation of the C1 complex which is composed of C1q, C1s and C1r. The formation of the complex is controlled by C1-inhibitor, which can physically interact with either C1s or C1r and thereby interfere with creation of the C1 complex. C1-inhibitor prevents the formation of the activation complexes of the lectin pathway and the thrombin pathway as well. *C1qa* −/− mice do not have functional C1 complex, and do not exhibit a neuronal migration defect. Therefore, we concluded that the classical pathway is not involved in the migration regulation. Our findings implicated *C1s* in regulation of neuronal migration; it is possible that C1s has activities that are beyond those related to the classical pathway. For example, it may be possible to postulate that knockdown of C1s may lead to higher levels of unoccupied C1-inhibitor that may in turn inhibit the lectin pathway, and this over-inhibition results in neuronal migration impairment. Unlike the process of synapse elimination, where the classical pathway plays a major role[4,5], our results point to the greater importance of the lectin pathway in the regulation of neuronal migration.

Mutations in components of the lectin pathway have been previously implicated in 3MC syndrome[9,10]. The 3MC syndrome is grouping of the Carnevale, Mingarelli, Malpuech and Michels syndromes, which are four rare autosomal recessive disorders[30–33] postulated to be part of the same clinical manifestation[34,35]. The main features of the 3MC syndrome are facial dysmorphic traits, including hypertelorism, blepharophimosis, blepharoptosis and highly arched eyebrows. Likewise, cognitive impairment, cleft lip and palate, postnatal growth deficiency and hearing loss are also consistent findings. Mutations were detected in two genes belonging to the lectin pathway; *COLEC11* encoding for the CL-K1 protein, which is a member of the protein family of C-type lectins (similar to MBL), *MASP1* encoding for the MASP1 protein, which binds MBL, and MASP3, a product of a differential splice form[9,10]. Our experiments using *Masp1* shRNA and *Masp1* knockout are expected to affect both gene products. Therefore, it is attractive to propose that intellectual impairment observed in most patients with 3MC syndrome may be explained in part by the role of the lectin pathway in the regulation of neuronal migration.

Consistent with this idea, database searches indicated that multiple complement pathway components are associated with ASD with different types of associations[36]. These disorders are typically characterized by social deficits, communication difficulties, stereotyped or repetitive behaviours and interests, and in some cases, cognitive delays. Neuronal migration deficits should be considered as one of the underlying pathologies in ASD[11]. New exome sequencing technology has identified novel rare variants and has found that sporadic cases of ASD are enriched for disruptive *de novo* mutations[37]. Therefore, we further analysed existing ASD exome sequencing data and show that ASD patients exhibit *de novo* mutations in complement genes[38–40] (Table 1). Mutations in 14 different genes were detected, additional evidence for the involvement of three of the genes in ASD exists (*COLEC12*, *C3* and *FGA*). Among the 16 identified mutations, 15 were analysed by functional prediction programs, 6 or 9 were found to be damaging or possibly damaging by SIFT or PolyPhen program, respectively. Nevertheless, it is likely that not only mutations that are predicted

to be damaging can be involved in disease, as has been recently demonstrated in patients with ASD or schizophrenia[41]. Underscoring our functional studies are damaging mutations detected in *C3* (K610T and D1569E) and in *MASP1* (Y294N). Collectively, our results and the abovementioned *de novo* mutations studies warrant further investigation to probe the involvement of the complement pathway in ASD.

## Methods

**Plasmids and primers.** The following shRNA were purchased from OpenBiosystems (ThermoScientific) in pLKO.1 vector: Masp1 shRNA (5′-CCTG TCCCTATGACTACATTA-3′); Masp2 shRNA (5′-CAGTCCCTTGTGACCA TTATT-3′); C3 shRNA (5′-GCCCGTGATTCACCAAGA AAT-3′); C1s shRNA (5′-GCAGCATACTACACTGCCAT-3′). The control shRNA plasmid used is pLKO.1-TRC control (Addgene) containing non-hairpin 18 bp sequence (5′-CCG CAGGTATGCAACGCG-3′). Plasmids containing the complete coding sequences of Masp1 (BC131638), Masp2 (BC013893.1) and C3 (BC043338.1) were purchased from OpenBiosystems (ThermoScientific) and subcloned into pCAGGS plasmid. These plasmids were used as a basis for creating shRNA resistant plasmids. Four to five mismatches with the shRNA sequence were inserted in the original sequences by PCR using the following primers: 5′-CCTGAGGTGCCCTGCCATATGA TTATATCAAGATTAAAG-3′ (Masp1); 5′-GGGAGACTCGGTACCATGCGAC CACTATTGCCACAACTAC-3′ (Masp2); 5′-GAGATGGGCCCGTAATCCA TCAGGAGATGATTGGTGGCT-3′ (C3). C3a was subcloned from the C3 plasmid with following primers: 5′-ATATGGCTAGCTCAGTACAGTTGATGGAAA-3′ and 5′- ATAGCGGCCGCTCACCTGGCCAGGCCCAGCACG-3′. C3b beta was subcloned from the C3 plasmid with following primers: 5′-ATATGGCTAGCA TCCCCATGTATTCCATCATT-3′ and 5′-ATAGCGGCCGCTCAGGCTG CTGGCTTGGTGCACTC-3′. The C3 fragments were subcloned into the pCAGGS plasmid that contained either C3 signal peptide (5′-ATGGGGGACCAGCTTCA GGGTCCCAGCTACTAGTGCTACTGCTGTTGGCCAGCTCCCCATTA GCTCTGGGG-3′) or CD33 signal peptide (5′-ATGCCGCTGCTGCTACTGCTG CCCCTGCTGTGGGCAGGGGCCCTGGCTATG-3′).

Lifeact-GFP was kindly provided by Dr Michael Davidson.
For Real-Time PCR experiments the following primers were used http://mouseprimerdepot.nci.nih.gov: 5′-AAGAACGGTTTACGGGCTTT-3′ and 5′-GGGACAGGAGCAGAAGTATCC-3′ for C1s; 5′-TATTGTGTCCCGG AGGGGT-3′ and 5′-CCAGCACCTGGTCTTCTGTT-3′ for Masp1; 5′-TTTGAG GCCTTCTATGCAGC-3′ and 5′-GCCCAAGTAGTTGTGGCAAT-3′ for Masp2; 5′-ATGCTGACCCTGAGGTCAAA-3′ and 5′-GGCCTTCTCTCTAACAGCCA-3′ for C3; 5′-ATTGGGACTGCTAGGCAATG-3′ and 5′-GGTGAGATGGAGGA ACCAGA-3′ for C3aR1; 5′-TTACCACAGAACCCAGGAGG-3′ and 5′-GCC ATCCGCAGGTATGTTAG-3′ for C5aR1.

**Antibodies.** The following antibodies were used for immunostainings: rabbit anti-C3 (Antibody Verify, 1:400, verified in this study, Supplementary Fig. 1a,b), rabbit anti-MASP1 (Antibody Verify, 1:100, verified in this study, Supplementary Fig. 1c,d), rabbit anti-MASP2 (Antibody Verify, 1:100, verified in this study, Supplementary Fig. 1e,f), chicken anti-TBR2 (Millipore, 1:100, AB15894), rabbit anti-TUJ1 (Abcam,1:1,000, ab18207), rabbit anti-CUX1 (anti-CDP, Santa Cruz, 1:100, SC-13,024), chicken anti-TBR1 (Millipore, 1:100, AB2261), rabbit anti-GFAP (DAKO, 1:500, Z0334), mouse anti-IdU-B44 (BD Biosciences, 1:200, 347580), rabbit anti-phospho-Histone H3 Ser10 (Millipore, 1:100), chicken anti-GFP (Abcam, 1:1,000), rabbit anti-Cleaved Caspase-3 (Asp175) (Cell Signaling, 1:200, #9661). Mouse CTR433 antibodies (1:50, Jasmin *et al.*, 1989, PNAS, 86: 2,051–2,055), a Golgi marker was kindly provided by Dr Michel Bornens (Institute Curie, Paris, France). Rat anti-C3b antibodies (Hycult Biotech, 1:500, HM1065) were used for western blot.

**Complement agonist peptides.** A selective C3aR agonist, WWGKKYR-ASKLGLAR ('super-agonist'[18]), and a C5aR agonist, YSFKPMPLaR ('EP54' (ref. 20)) were synthesized as previously described[18,20]. It should be noted that the C5aR agonist also activates C3aR (ref. 19) and thus is described herein as a dual C3aR/C5aR agonist. The agonists ($1 \mu g \, mg^{-1}$) were injected to the ventricles of the embryos together with the indicated plasmids.

**Animals.** Animal protocols were approved by the Weizmann Institute IACUC and were carried out in accordance with their approved guidelines. ICR mice were purchased from Harlan laboratories. C1qa-deficient mice (*C1qa* KO) were kindly provided by Prof. Marina Botto[15]. These mice were created by targeting the neomycin-resistance gene in exon 1 of C1qa and were backcrossed to C57BL/6 background. *C3* KO mice were obtained from The Jackson Laboratory. Male and female embryos were used in the study.

**CRISPR/Cas9 knockout generation.** Cas9 plasmid and plasmids encoding guide RNAs were purchased from the University of Utah Mutation Generation lab. The

following oligos were used for construction of gRNA vectors: Masp1: 5′-ACACC GCTTGATTCGAAACCCCTCCG-3′ and 5′-AAAACGGAGGGGTTTCGAAT CAAGCG-3′ (location 5:50012720–50012743: − strand); Masp2: 5′-ACACCG ACACCAGGCGCCCGAATACG-3′ and 5′-AAAACGTATTCGGGCGCCTGGT GTCG-3′ (location 18:50048074–50048097: + strand).

*In vitro* transcribed Cas9 RNA($100 \, ng \, \mu l^{-1}$) and sg RNA($50 \, ng \, \mu l^{-1}$) were injected into one cell fertilized embryos isolated from superovulated CB6F1 hybrid mice mated with CB6F1 males Harlan Biotech Israel Ltd. (Rehovot, Israel). Injected embryos were transferred into the oviducts of pseudopregnant ICR females as previously described[42]. At day E14.5 the pregnant mice were subjected to IdU injection. Genomic DNA from the treated embryos was analysed for mutations in the mutated genes using High Resolution Melt analysis and confirmed by Sanger sequencing.

**IdU injections.** The thymidine analog iododeoxyuridine (IdU) was injected intraperitoneally (0.01 ml of $5 \, mg \, ml^{-1}$ IdU solution per gram body weight) into pregnant mice at the indicated time points.

***In utero* electroporation.** Plasmids were transfected by *in utero* electroporation using previously described methods[43,44]. Before the surgery the animals were injected with buprenorphine ($2 \, mg \, kg^{-1}$ BW, subcutaneously). Pregnant ICR mice at E14.5 days post gestation (E14) were anaesthetized by injection of ketamine 10%/xylazine $20 \, mg \, ml^{-1}$ (1/10 mixture, 0.01 μl per g of body weight, intraperitoneally), alternatively Isofluran anaesthesia was used. The uterine horns were exposed, and plasmid mixed with Fast Green ($2 \, \mu g \, \mu l^{-1}$, Sigma) were microinjected through the uterus into the lateral ventricles of embryos by pulled glass capillaries (Sutter Instrument, Novato, CA, USA). The concentration of plasmids was $0.5 \, \mu g \, \mu l^{-1}$ for pCAGGS-GFP, $2 \, \mu g \, \mu l^{-1}$ for shRNA construct and 1–1.5 μg for overexpression plasmids. Electroporation was accomplished by discharging five 41 mV 50 ms long pulses with 950 ms intervals, generated by a NepaGene electroporator. The pulses were delivered using 10 mm diameter platinum plated tweezers electrodes (Protech international Inc., San Antonio, TX, USA) placed at either side or the head of each through the uterus. Animals were killed 4 days after electroporation at E18.5 (E18). Embryos with well-distinctive positive GFP signal in cortex visible through fluorescent binocular were intracardially perfused using 4% paraformaldehyde-phosphate buffered saline (PFA-PBS). Embryos with dotted, double hit or hit outside the cortex were not included in the study. Brains were post-fixed overnight and sectioned (60 μm; vibratome, Leica). For examination of long-term effects of the treatments *in utero* electroporation was performed at E14, the mice delivered and the pups of postnatal day 8 (P8) were used for the experiments. For proliferation experiments embryos were *in utero* electroporated on E13.5 (E13) with 7 mm electrodes (39 mV pulses). IdU was injected in 24 h after electroporation for 30 min. Post-fixed brains were cryopreserved in sucrose and cryosections (10 μm) were used for proliferation analysis. *In utero* electroporation of C1qa KO mice and C3 KO mice were performed at E14 with 7 mm electrodes (39 mV pulses). C57BL/6 WT mice were used as a control for C1qa and C3 KO mice.

**Immunocytochemistry.** Floating vibratome sections (60 μm) were permeabilized using 0.1% Triton X-100 and blocked in blocking solution (PBS, 0.1% Triton X-100, 10% HS; 10% FBS) for 60 min. Antibodies were incubated in blocking solution overnight at 4 °C. After washing, appropriate secondary antibodies (Jackson ImmunoResearch) were diluted in blocking solution, and incubated for 2 h at room temperature. Slices were mounted onto glass slides using Aqua Polymount (Polysciences). For IdU immunostainings (E18) the brain slices were pretreated with HCl (30′) followed by neutralization with borate buffer. For IdU immunostainings (E13 and E14), the cryosections (10 μm) were used. Antigen retrieval procedure was performed by boiling slides in sodium citrate buffer (10 mM, pH6) for 30 min.

**Microscopy, quantification and statistical analyses.** Images were taken using either Pannoramic MIDI scanner (3DHisthech) or confocal microscopy (LSM780, LSM800 Zeiss), equipped with Axio Observer Z1 microscope, and imaged with either Plan-apochromat ×20/0.8, or Plan-apochromat ×40/1.2, or Plan-apochromat ×63/1.4 oil objectives. The scaling data are 0.624 × 0.624 μm per pixel for ×20 magnification, 0.312 × 0.312 μm per pixel for ×40 magnification, and 0.198 × 0.198 × 0.51 μm per voxol for ×60 magnification. The images were processed by ZEN software and/or Imaris software.

Cell count, positioning and colocalization analyses were performed using Imaris software (Bitplane Inc., Zurich, Switzerland, Imaris core module). At least three brains were analysed for each treatment. Four representative slices from each brain were chosen for analysis. The size of the area of interest was determined and preserved per each experiment. For each slice the area of interest was positioned so that the centre of the electroporated area is in the centre of the area of interest. For the cell count and positioning the relevant channel of an area of interest was analysed with 'Spots' module of Imaris, every spot labelling approximate centre of the cell body. The 'y' position of all the dots was analysed by Microsoft Excel Histogram tool. Statistical analysis was performed by one-way or two-way analysis of variance followed by Bonferroni multiple comparison analysis, using

PRISM 7 for Mac (GraphPad software). Error bars represent standard error. For the measurement of fluorescent signal, high-resolution z-stack images were collected of the relevant immunostained slices. The analysis was performed with ImageJ program. The outline of the cells marked with Lifeact GFP served for the freehand selection of the individual cell. The area and integrated density of the relevant channel for each cell was measured, and CTCF (corrected total cell fluorescence) was calculated as Integrated Density − (Area of selected cell × Mean fluorescence of background readings). The data were corrected by the cell area parameter.

**Real-time qRT–PCR.** For confirmation of shRNA efficiency neurospheres from E13.5 were grown in Neurobasal medium (Gibco) supplemented with B27, glutamax, gentamicine, EGF (20 ng ml$^{-1}$), bFGF (20 ng ml$^{-1}$) and heparin for 2 days. The cells were transfected by NEPA21 electroporator (Nepagene) according to the manufacturer's instructions. The cells were grown for additional 48 h and collected for RNA isolation (TRI reagent, Sigma). After Dnase treatment (Sigma), first-strand cDNA synthesis was done using M-MLV RT (Promega). Relative levels of genes of interest expression were normalized to the 29 rps gene. Real-time PCR with SYBR FAST ABI qPCR kit (Kapa Biosystems) was performed using StepOnePlus Real-Time PCR System (Applied Biosystems). E13, E14, E16, E18 cortices ($n = 3$ for each time point), and liver tissue was dissected in cold PBS and fast-freeze in liquid nitrogen. RNA preparation and Real-time RT–PCR were performed as described above.

**ELISA.** The Complement Fragment 3a (C3a) BioAssay ELISA Kit (mouse) was purchased from US Biological. The cortex from E16 embryos were dissected and homogenized in cold PBS following two freeze-thaw cycles to break the cell membranes (350 μl of PBS were used for two cortices of each brain). The homogenates were centrifuged and the supernatant used for the assay and performed following the company instructions in 1:10 dilution. Wild-type embryos ($n = 2$) were used to determine the amount of C3a peptide in the cortex. Littermate C3 KO embryos ($n = 2$) were used as a negative control. Two technical repeats were performed. The experiment was performed twice.

**Western blot analysis.** The cortices from E13, E14, E16 and E18 embryos were dissected and homogenized in lysis buffer (50 mM Tris-HCl pH7.5; 150 mM NaCl; 1 mM EDTA; 1 mM EGTA; 1% Triton X-100) supplemented with protease inhibitor cocktail (Sigma). Thirty micrograms of total protein were mixed with SDS sample buffer, separated by SDS–PAGE and subjected to western blot analysis with the indicated antibodies. Images (Supplementary Fig. 1n) have been cropped for presentation. Full size images are presented in Supplementary Fig. 8.

**Data availability.** All relevant data are available from the authors.

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

## Acknowledgements

We are grateful for the help of Golda Damari, Alina Maizenberg, Sima Peretz, Ofira Higfa, Yehuda Melamed, Osnat Amram, Oz Yirmiyahu, Talia Levy, Vanessa Zamor, Sergey Viukov, and all lab members from the Weizmann Institute of Science, Allison R. Bialas and Beth Stevens from the Children's Hospital, Harvard Medical School, Boston, Marina Botto from Imperial College London, and Timothy Dahlem, Utah University. O.R. is the incumbent of the Bernstein-Mason Chair of Neurochemistry. The research has been supported by the Israel Science Foundation (grant no. 347/15), the Legacy Heritage Biomedical Program of the Israel Science Foundation (grant no. 322/13), Weizmann-FAPESP supported by a research grant from the Dr Beth Rom-Rymer Stem Cell Research Fund, Nella and Leon Benoziyo Center for Neurological Diseases, Yeda-Sela Center for Basic Research, Jeanne and Joseph Nissim Foundation for Life Sciences Research, Wohl Biology Endowment Fund, Fritz Thyssen Stiftung, Lulu P. & David J. Levidow Fund for Alzheimer's Diseases and Neuroscience Research, the Helen and Martin Kimmel Stem Cell Research Institute, the David and Fela Shapell Family Center for Genetic Disorders Research. T.M.W. is supported by a National Health and Medical Research Council Career Development Fellowship (APP1105420).

## Author contributions

A.G., T.S., and R.H.-K. planned and conducted the experiments, analysed the data and wrote the manuscript. T.O. performed bioinformatics analysis and wrote the manuscript. T.M.W. contributed the agonists and wrote the manuscript, and O.R. planned the experiments, analysed the data and wrote the manuscript.

## Additional information

**Competing interests:** The authors declare no competing financial interests.

