## [Peer Review File · Nature Communications]

Reviewers' comments:

Reviewer #1 (Remarks to the Author):

The manuscript by Gorelik et al., entitled "Unexpected Activities of the Complement Pathway in Migrating Neurons", provides evidence that proteins in the lectin arm of the complement pathway are expressed during development and play a role in neuronal migration. Overall, the manuscript is high quality and significant. The data are noteworthy and provide new insights into the role of immune molecules in brain development. There are some remaining concerns regarding important control/validation experiments that need to be addressed. In addition, the link with ASD is tenuous and omission of these data should be considered. See detailed major and minor concerns below.

Major concerns:

1. The connection with autism spectrum disorders and schizophrenia is very weak. It is unclear how the *in silico* analysis of gene expression in ASD patients correlates with data presented in this paper. Given that the complement pathway has also been linked to synaptic pruning (Stevens et al. 2007, Schafer et al. 2012), it is difficult to make a direct link between neuronal migration phenotypes in the mice and ASD. It is recommended that these data be omitted from the revised version. To improve the correlation with ASD and add functional data to the paper, behavioral deficits in C3, Masp1, or Masp2 knockout animals would help. However, the neuronal migration link would still only be correlative and the synaptic pruning defect is still an underlying cellular defect that may also affect behaviors.

2. The authors never address what cell types express C3, Masp1, Masp2, C5aR, or C3aR. Given the timing of electroporation experiments, manipulations should primarily manipulate excitatory neurons but this is never validated or discussed in results. In addition, protein localization in Figure 1 suggesting protein is extracellular is difficult to visualize and requires increased magnification and 3D rendering/super resolution imaging. It is also curious that the authors show intracellular localization of protein in Figure S2.

3. It is unclear why the authors switched from using Masp2 and Masp1 KO mice in figure 1 to shRNA knockdown in subsequent figures. It is assumed that the KO mice are embryonic lethal but the authors never address this explicitly. In addition, neuronal migration in representative figures is highly variable in Masp1 and Masp2 KO mice versus Masp1 and Masp2 shRNA. For example, in Masp1 and Masp2 KO mice there is migration to the cortical plate; however, there is no migration to cortical plate with shRNA knockdown.

4. The authors should quantify overall neuron cell numbers in cortical plate versus ventricular zone, particularly in KO mice. If there is a migration defect, one would expect a decrease in neuronal cell numbers in cortical plate over development and in adult mice. Similarly, the authors should demonstrate co-localization with neuronal markers in postnatal animals (Figure 2 I-J).

5. For electroporation experiments, the authors need to demonstrate that efficiency is similar across experiments.

6. Most validation of shRNA and agonists has not been done or is not high quality.

a. The authors state that C3, Masp1, and Masp2 are localized extracellularly in figure 1, however have one zoomed in image of intracellular localization to validate shRNA in supplementary figure 2. The authors need to quantify these data blindly. A better experiment would be to FACS GFP-positive cells and perform qPCR and western blot to quantify knock-down.

b. Validation and dose response of the C5aR and C3aR agonist needs to be completed.

7. There is a general lack of experimental detail in the materials and methods section. In particular, description of imaging and quantification is sparse. The authors need to elaborate on how the images were acquired, z-stacks, step size, and how images were quantified. Furthermore, the authors do not describe when C5aR and C3aR agonists were given or at what dose.

8. The data suggests that C1qa knockout doesn't cause defects in neuronal migration, however C1s

knockdown does show a neuronal migration defect. C1s is part of a complex with C1qa, this discrepancy needs to be addressed.

Minor concerns:

1. The authors use different developmental time points to demonstrate that C3, Masp1, and Masp2 are expressed during development in figure 1. This is curious and difficult to compare molecule expression patterns. Furthermore, this is not clear until one reads the legend carefully.
2. In Figures 2-4, it is unclear how cells are labeled for quantification. It is assumed that the authors are showing GFP labeled cells following electroporation and taking a % of GFP positive cells but this needs to be stated.
3. In Figure 4 d and h, the authors do not include control shRNA controls on the graphs.
4. The authors show in situ hybridizations in supplemental figures 1 and 2. These data do not add much to the paper and are from an online database. They also do not have the proper KO controls, which are important for interpreting in situ.
5. Authors state that they used two-way ANOVAs throughout the study. There are clear cases where one-way ANOVAs are more appropriate (e.g., Figures 2&3).
6. In Supplemental Figure 1, the authors include a western blot to show C3b. These data are weak and it is unclear why other C3 fragments are not also visualized. A clear positive control (such as serum) included on the blot would help.
7. Data referenced in that paper, such as ELISA and qPCR, are not shown. Please include these data.
8. In the discussion, the authors claim, "The most persistent impairment of neuronal migration was observed in the case of Masp1 shRNA, where at P8, treated neurons did not reach the cortical plate and acquired a fate of deep layers", yet they do not show P8 data for any other manipulation. In addition, pharmacological activation of the C3a and the C5a receptor in Masp1 shRNA knockdown to ameliorate these effects, they only show data for C3 and Masp2 knockdown.
9. In the discussion, the authors write "Knockout of lectin pathway components resulted in stronger migration phenotypes". Stronger than what? It is assumed they are referring to the classical pathway; however, there does seem to be a significant defect with C1s shRNA.
10. The writing in the abstract, introduction, and discussion is often disjointed. The authors should revise to increase readability and fluidity.

Reviewer #2 (Remarks to the Author):

The manuscript by Gorelik et al describes that some complement components are required for radial migration of cortical neurons. The authors first showed the expression pattern of the complement components, C3, MASP1 and MASP2 in developing mouse brains. Second, using in utero electroporation, the authors analyzed the consequences of either knockout or knockdown of key component of two branches of the complement cascade, the lectin and the classical pathway, on radial migration. They found that while the lectin pathway is likely regulating neuronal placement, the classical pathway is mildly involved. By analyzing the position of neuronal cell at later stages (P8), the authors showed that MASP1 depletion not only permanently impairs neuronal placement but also affects the specification of the misplaced neurons. Third, Gorelik and colleagues showed that both C3 cleavage and increased activation of the downstream receptors of the cascade (C5aR and C3aR) rescued the migration deficit observed in absence of MASP2 or C3. The authors finally identified de novo mutation in complement genes in ASD patients and, therefore, suggested an involvement of the complement pathway in ASD etiology.

Even though the authors do not propose any mechanism by which the lectin pathway could regulate neuronal migration, they provide strong evidences for a function of this complement pathway in the embryonic brain. However, the authors need to address a number of concerns to substantiate their

conclusion.

Major points

1) As C3 and *masp1* and *masp2* seems to be expressed in progenitors as well, how the authors are sure that the phenotype they observed is strictly due to migration defect and not to additional defect occurring in progenitors? Authors should moderate their conclusion and talk about positioning instead of migration; Induction of shRNA specifically in postmitotic neurons by in utero electroporation of CRE-inducible shRNA (for *Masp1* or *Masp2*) should determine whether the abnormal migration results or not from defect in progenitors (see Garcez et al, nature communication, 2015).

2) Could the author discuss why the Migration/Position of *Masp1* KD cells is only partially rescued by ectopic expression of *Masp1*? Same comment for figure 3b-g (C3 shRNA) and figure 3l-q (*masp2* shRNA). This partial rescue could reflect off targeting of the shRNAs (of note, many shRNA (even ct shRNA) are shown to give off target effect by buffering endogenous miRNAs machinery (Baek et al, neuron, 2014)). Does expression of *Masp1* rescue the positioning of GFP cells in *Masp1* KO mice? On the same line, while the C3aR/C5aR agonist only partially rescues the phenotype upon C3 KD (fig 4e-h), it seems to completely rescue the phenotype in C3KO (fig 4i-k). (see also minor point 21)

3) The authors dismissed the role of the classical pathway in regulating neuronal positioning based on two results: 1/ no phenotype is observed in C1qa KO mice (fig 1 o-q); 2/ only a mild phenotype is observed upon C1s depletion (fig S1e-h). However:

- They haven't validated the complete depletion of C1qa in the C1qa KO brains.

- C1s shRNA reduces C1s by 48% in comparison with control shRNA. One cannot exclude to get a more striking phenotype using more efficient shRNAs. An alternative could be to use siRNAs.

4) *Masp1* KD leads to permanent arrest of cells in deep layers. Could the authors comment on the following points:

- From pictures shown in figure 6i-j, it seems that ALL the *Masp1* depleted cells are arrested in deep layers. Are those images representative? Would some cells still reach the CP or be dispersed throughout the CP? If yes, please provide more accurate pictures and quantification. If not, what are becoming the cells that migrate out the IZ at E18 (2c)? Do you observe any cell death? In the same line, if some cells reach the upper layers, are those cells well specified (*cux1*)?

- Do the authors observed defect in layering in *masp1* KO mice? Immunostaining of P2 or P8 *Masp1* KO mice with layer specific markers should clarify this point.

- Figure 2k-p: please provide pictures encompassing the entire cortical thickness.

- immunostaining with upper layer markers (*cux1*, *satb2*) should be included to make sure that all the stalled cells are not correctly specified. If required, provide quantification.

5) The authors showed a defect of specification. This defect can explain the impaired positioning of cortical neurons. One can imagine that the migration process by itself (multi-bipolar shape conversion, locomotion) is not affected. Could the author clarify what are the root causes of the phenotype?

6) To analyse migration, the authors use in utero electroporation of GFP except in *Masp1* and *Masp2* KO mice where they use IdU birthdating (figure 1s-w). Could the authors explain why they are not using the same technique in every KO models? In utero electroporation of GFP plasmid in *Masp1* and *masp2* KO mice would greatly help to compare the different sets of experiments.

7) The authors showed that reduction by only 38 % of *Masp1* induces a strong migration phenotype with very few depleted cells reaching the cortical plate (12%) compared to the control (figure 2a-d), while complete deletion leads to a milder phenotype (30% of cells in CP)(figure 1 u-w). Could the authors discuss this particular point?

8) *Masp2*, the closest paralog of *Masp1* partially rescued the phenotype in *Masp1* KD neurons (Fig 2g-h), suggesting functional overlap between those two proteins. One protein could compensate for the loss of the other in vivo. Could this explain the mild phenotype in *masp1* or *masp2* KO mice compared to C3 KO mice? To test this hypothesis, authors could perform GFP in utero electroporation in *masp1/masp2* double KO and test if they worsen the phenotype compared to single KO. Are *masp1* or *masp2* more expressed in *masp2* or *masp1* KO brains, respectively?

9) In Figure 1, the authors analyzed the expression of C3, Masp1 and Masp2 in the developing cortex. While C3 expression/localization has been assessed at E16.5, the masp1 and masp2 immunostainings have been performed at E14.5 and E18.5. Could the authors explain their choices? Does that mean that C3 is not or slightly expressed at later time points? It would have been more informative to: 1) analyze by western blot (or alternatively qPCR) the amount of each protein at those different developmental stages to test whether their expression level change over the time; and 2) show pictures of entire thickness of the cortex at each stage instead of magnification at E18.5. On the same line, one doesn't know which part of the cortex (CP, VZ, IZ) is shown in 1e and 1h.

10) Figure S1A shows a decreased cleavage of C3 as the development progresses. Could the authors comment about a possible reduction of complement activity throughout the development? Could this decrease result from a reduction in C3 protein level?

11) As C3 cleavage products only partially rescue masp2 KD phenotype, could the authors discuss other possible function of masp2?

12) As the entire rescue experiments are performed in masp2 shRNA electroporated cells, why do the authors analyze postnatal consequences after masp1 KD in figure 2? It would have been better to perform those analyses in the same model.

Minor points

13) While the text indicates that the immunostainings of C3 have been performed at both E14,5 and E18,5, the figure legend mentions E16,5.

14) Figure 1 legend: could the authors homogenize the developing stages (E14,5 or E14, E18,5 or E18)

15) Figure 1e-g: the authors claim that "high levels (of masp1) were noted in the VZ, upper IZ and upper CP. Colocalisation with specific markers for those region will strengthen this conclusion.

16) Masp1 and Masp2 KO mice: although those models have been used only in figure 1, a full validation of those models engineered using the CRIPR/Cas9 technology is missing. Has the authors assessed possibility of off targets?

17) Supplemental document: IdU injections section: BrDU is mentioned??

18) For greater clarity, S1f and h should be merged (as presented in figure 4). Same comment for figure 2b-d-f -h; 3-c-e-g-l-k and 3m-o-q-s-u. It will help comparisons.

19) The authors show statistics in comparison to shRNA condition. Adding statistics in comparison to ct condition would help to decipher between total or partial rescue. (figure 2b-d-f -h; 3-c-e-g-l-k and 3m-o-q-s-u)

20) As control, the authors performed GFP or Ctl shRNA in utero electroporation in WT mice. Could the authors comment the difference of migration profile in those two types of control (70 % and 50% of cells reaching the CP after GFP or shRNA electroporation, respectively)? Could the authors clarify in which mouse background in utero electroporations of shRNA have been performed?

21) For greater clarity, authors should include ct condition in their graph (figure 4).

Reviewer #3 (Remarks to the Author):

In this manuscript, Gorelick et al clearly demonstrate that activation of the lectin arm of the complement pathway regulates neuronal migration. These studies not only provide new insights into the mechanisms of neuronal migration, but also underscore the relevance of this cortical developmental process to ASD. It shows that immune system malfunctions during development may exert much wider effect on brain development than previously thought. The authors have used comprehensive developmental expression, gene knockdown/ knock out studies of complement pathway genes (C3, Masp1, 2) to illustrate this. In general, this study is logically organized, clearly illustrated, and provides much needed new information on immune system-brain development cross talk. Addressing the following points will be necessary to strengthen the clarity and impact of this

work.

- 1) While its clear that C3/ Masp 1, 2 knockout or knock down affects neuronal migration, some of the reduced neuronal numbers/placement effects may have ben due to altered proliferation of cortical progenitors in these experimental models. Please provide evidence to rule out this possibility.
- 2) In C3 and Masp1, 2 null mice, please show altered migration/ placement with standard cortical layer specific neuronal markers (partially addressed in Fig.2, but not adequately).
- 3) Does activation of the lectin arm of complement pathway cause death of migrating neurons?
- 4) While its admittedly beyond the scope of this work, do the authors have any evidence to show that the damaging mutations detected in C3 (K610T and D1569E) and in MASP1 (Y294N) affects neuronal migration. Such information will provide direct evidence for the connection between complement pathway, neuronal migration, and ASD, currently missing from this work.

We want to thank the reviewers for carefully going over our manuscript and providing detailed comments. Our manuscript was completely re-written, and the figures and supplementary figures were revised as well. Hereby, we addressed the comments in a point-to-point manner. We do hope that at the present time, our manuscript is suitable for publication in *Nature Communications*.

Reviewers' comments:

Reviewer #1 (Remarks to the Author):

The manuscript by Gorelik et al., entitled "Unexpected Activities of the Complement Pathway in Migrating Neurons", provides evidence that proteins in the lectin arm of the complement pathway are expressed during development and play a role in neuronal migration. Overall, the manuscript is high quality and significant. The data are noteworthy and provide new insights into the role of immune molecules in brain development. There are some remaining concerns regarding important control/validation experiments that need to be addressed. In addition, the link with ASD is tenuous and omission of these data should be considered. See detailed major and minor concerns below.

Major concerns:

1. The connection with autism spectrum disorders and schizophrenia is very weak. It is unclear how the *in silico* analysis of gene expression in ASD patients correlates with data presented in this paper. Given that the complement pathway has also been linked to synaptic pruning (Stevens et al. 2007, Schafer et al. 2012), it is difficult to make a direct link between neuronal migration phenotypes in the mice and ASD. It is recommended that these data be omitted from the revised version. To improve the correlation with ASD and add functional data to the paper, behavioral deficits in C3, Masp1, or Masp2 knockout animals would help. However, the neuronal migration link would still only be correlative and the synaptic pruning defect is still an underlying cellular defect that may also affect behaviors.

We are moving the connection with ASD to the discussion, which may have some room for speculations.

2. The authors never address what cell types express C3, Masp1, Masp2, C5aR, or C3aR. Given the timing of electroporation experiments, manipulations should primarily manipulate excitatory neurons but this is never validated or discussed in results.

The experiments were performed on E14 mouse embryos. These experiments are indeed targeting cells in the VZ of the developing cortex, that are destined to become upper layer excitatory neurons.

In the immunostainings for C3, MASP1 and MASP2, we added additional markers, which label different areas in the developing cortex.

Added to the text:

“Immunostaining demonstrated wide-spread immunoreactivity of C3, MASP1 and MASP2 in embryonic brain sections (E16, Fig. 1b-d). Relatively low level of expression was noted in the ventricular zone, and the subventricular zone, where Tbr2 positive cells reside (Fig. 1e). Expression of C3, MASP1 and MASP2 was noted in areas where postmitotic neurons are present (Tuj1 positive, Fig. 1f), and higher levels of expression were detected in the cortical plate, where Tbr1 immunostaining marks the deep layers (Fig. 1g).”

We validated the identity of the target cells by immunostainings with anti-Cux1 antibodies (figure 2 k-n). These immunostainings demonstrate that all the electroporated cells (GFP positive) become Cux1 positive neurons.

Added in the text:

“Next, the postnatal effects of these embryonic interventions were studied. Following C3 shRNA treatment of the embryonic brains, most of the neurons reached the superficial layers at postnatal age, whereas in case of either *Masp1* or *Masp2* shRNA a significant proportion of the cells did not reach their proper position. Many of the *Masp2* shRNA cells were positioned ectopically in the white matter. *Masp1* shRNA treatment resulted in cells abnormally positioned in the IZ/CP border (Fig. 2k-r). In addition, a difference in the relative positioning of the Cux1 positive cells in the CP was noted. C3 and *Masp2* knockdown resulted in a more scattered, less compact Cux1 layers. A small proportion of the *Masp1* shRNA treated cells exhibited an over-migration phenotype and Cux1 positive cells were found in the upper layer I. To confirm the fate of the cells in ectopic positions we have used antibodies for the upper layer marker Cux1 and the deep layer marker Tbr1. Cux1 positive cells in the Control shRNA treated brain were occupying the superficial layers II-IV (Fig. 2k). Interestingly, the stalled C3, *Masp1* and *Masp2* shRNA treated cells were Cux1 positive (Fig. 2l-n). Both control and shRNA treated cells were negative for the deep layer neuronal marker Tbr1 (Fig. 2 o-r). In addition, the *in utero* electroporated cells were negative for the glial marker GFAP (supplementary Fig. 2g-j).”

In addition, protein localization in Figure 1 suggesting protein is extracellular is difficult to visualize and requires increased magnification and 3D rendering/super resolution imaging. It is also curious that the authors show intracellular localization of protein in Figure S2.

We followed this advice and have conducted new experiments. To clearly label the cell boundaries, we *in utero* electroporated E14 embryos with Lifeact GFP and fixed the brains at E16. Vibrotome sections were immunostained with the respective antibodies and images 3D images were reconstructed using confocal microscopy and IMARIS software. In the modified figure 1 (Fig 1 i-k), we show low and high magnification images. The high magnification images included the respective Z-images, which clearly demonstrate the extracellular localization of the complement proteins examined in this study.

Added in the text:

“To better visualize the cellular localization of the complement proteins, E14 mouse embryos were *in utero* electroporated with Lifeact-GFP that marks the cell

periphery, and the brain sections were immunostained at E16. Neuronal progenitors that are electroporated in the ventricular zone (VZ) will differentiate and follow the normal migratory path to superficial layers of the cortical plate (CP) at E18 (Fig. 1h). At E16 the majority of migrating neurons electroporated on E14 undergoes multipolar-to bipolar transition in the intermediate zone (IZ). High resolution and Z-stack images of neurons taken from the intermediate zone (relative position marker by * in Fig. 1b-d) demonstrate that most of the complement protein deposits are extracellular (Fig. 1i-k)."

Figure S2 (a-f) demonstrates neurons in culture, these cells demonstrate intracellular localization of a protein that is destined for secretion (i.e. Golgi), it is not possible to demonstrate by immunostainings extracellular localization of proteins in sparse 2D culture. The intracellular proteins visualized in the figure are most likely undergoing synthesis and/or secretion and/or following internalization. In cell culture there is not enough extracellular matrix to accumulate the secreted protein.

3. It is unclear why the authors switched from using Masp2 and Masp1 KO mice in figure 1 to shRNA knockdown in subsequent figures. It is assumed that the KO mice are embryonic lethal but the authors never address this explicitly. In addition, neuronal migration in representative figures is highly variable in Masp1 and Masp2 KO mice versus Masp1 and Masp2 shRNA. For example, in Masp1 and Masp2 KO mice there is migration to the cortical plate; however, there is no migration to cortical plate with shRNA knockdown.

We did not generate Masp2 and Masp1 KO lines, we analysed mutant embryos, each of which was a result of a single CRISPR/Cas9 injection in the fertilized oocyte. We agree that the KO and knockdown are not identical, however they do share common features. In most instances where KO and knockdown are studied in parallel, the knockdown phenotypes are more pronounced than the KO. This can be possibly explained by gene redundancy, which can occur in the KO but not the acute knockdown. A convincing and comprehensive study was performed on this topic ¹.

Alternatively, or in addition, possible changes in the observed phenotype may be due to different techniques used in the analysis. The knockdown and knockout are analysed in two different and widely used methods. Whereas in case of KD in utero electroporation studies were performed in case of the KO cells were labelled by IdU. GFP in utero electroporation usually results in a more uniform label of cells, whereas in case of IdU it is possible that we are analysing also cells that underwent one cell division. It is difficult to distinguish between the first and second generation of cells and this results in a more dispersed presentation of the cells. Even though the techniques differ, we do not claim that the phenotypes are the same.

Historically, we initiated our studies using knock-down systems because:

- a) the experiments with ICR mice require less animals to be sacrificed for each experiment with adequate results compared to C57/Bl6 mice.

- b) Using knock-down we can introduce several experimental combinations (knock-down, cell labelling, rescue constructs).
- c) Especially when we are analysing a pathway, which involves multiple genes, knock-down systems are much more time-efficient versus the time required to generate multiple knock-out genes (especially when we are discussing the pre-CRISPR/Cas9 era).
- d) The only mouse line that was available for our studies was C3 knockout.
- e) We generated Masp1 and Masp2 knock-out using CRISPR/Cas9 gene editing technology and analysed embryos directly. This required injecting the pregnant mice with BrdU or IdU and using all the embryos generated by this technique, genotyping, and immunostaining. Each embryo is an independent targeted event and several embryos were analysed in each case. We did not keep the lines.

4. The authors should quantify overall neuron cell numbers in cortical plate versus ventricular zone, particularly in KO mice. If there is a migration defect, one would expect a decrease in neuronal cell numbers in cortical plate over development and in adult mice. Similarly, the authors should demonstrate co-localization with neuronal markers in postnatal animals (Figure 2 I-J).

We quantified the labelled neurons in entire width of the cortex in the KO (either GFP labelled in the C3 KO or IdU labelled in the Masp1 and Masp2 KO), we analysed the relative position of the labelled cells (Figure 1 I-t). In case of a migration defect, we expect that a smaller proportion of the labelled cells will reach the superficial layers of the cortical plate. In addition, we immunostained the cortices with anti-Cux1 and anti-Tbr1 antibodies and noted significant changes in the width of immuno-positive domains (Figure 1 u-af). We added images of immunostainings and co-localizations with neuronal marker in postnatal animals (Figure 2 k-r).

5. For electroporation experiments, the authors need to demonstrate that efficiency is similar across experiments.

In our experimental system, we scan the *in utero* electroporated brains by fluorescent microscopy, therefore, we exclude brains in which only a small number of cells are GFP positive. All the results are normalized to the total number of GFP positive cells in the slice.

6. Most validation of shRNA and agonists has not been done or is not high quality.
a. The authors state that C3, Masp1, and Masp2 are localized extracellularly in figure 1, however have one zoomed in image of intracellular localization to validate shRNA in supplementary figure 2. The authors need to quantify these data blindly. A better

experiment would be to FACS GFP-positive cells and perform qPCR and western blot to quantify knock-down.

The cells in supplementary Figure 2 are neurons in culture and as stated above, these cells do show intracellular staining of C3, MASP1 and MASP2. In addition, all shRNA in the experiments were validated for qPCR from transfected neurospheres. We estimate that the transfection efficiency was no more than 50%. Therefore, the reduction we see with the qPCR is significant.

We tested multiple antibodies, and we were not successful in finding antibodies that work well in Western blots.

b. Validation and dose response of the C5aR and C3aR agonist needs to be completed.

The C3aR/C5aR agonists used in our study have been previously validated pharmacologically. The C3a agonist, WWGKKYRASKLGLAR, is a modified fragment of active region of the C3a protein, and exerts potent activity at the C3a receptor. It was originally described² and has been characterised in several pharmacology studies since. For example, a study by our co-author, TMW, showed its potency (EC50) to be ~200nM³, and that WWGKKYRASKLGLAR was active on mouse cells, but completely inactive in C3aR knockout cells, demonstrating its selectivity for C3aR⁴.

The dual C3a/C5a agonist YSFKPMPLaR, was also designed as a modified fragment of the active region of the C5a protein, and was originally described as a potent C5a agonist⁵. Studies since then however, have demonstrated that this agonist also binds with high affinity to the C3aR, and induces functional responses with an EC50 of ~200nM⁶. Hence it is now preferentially referred to as a dual C3a/C5a agonist.

Given these prior findings, for the current study, we utilised WWGKKYRASKLGLAR as a potent and selective C3a agonist, and YSFKPMPLaR as a potent dual agonist of both C3aR and C5aR, and selected the doses of 1 µg /mg. Regardless, to further confirm their specificity in the current study, we provide here a figure for the reviewer (figure R1) showing that half of the dosage of the agonist shows reduced rescue activity.

Figure R1. Brains were electroporated in utero (E14-E18) with Masp2 shRNA (a), or Masp2 shRNA together with the dual agonist (b). The same experiment was performed with half dosage of the agonist (d). As evident, less GFP+ cells reached the cortical plate when the lower dosage was used, suggesting this concentration was less effective (compare b and d). The scale bar is 50 μm .

7. There is a general lack of experimental detail in the materials and methods section. In particular, description of imaging and quantification is sparse. The authors need to elaborate on how the images were acquired, z-stacks, step size, and how images were quantified. Furthermore, the authors do not describe when C5aR and C3aR agonists were given or at what dose.

The dosage and time for agonists injection are added to the “complement agonist peptides” section of the Methods. Additional details about acquisition and analysis of images were included into the “Microscopy, quantification and statistical analyses” section.

Added in the text:

Complement agonist peptides

A selective C3aR agonist, WWGKKYRASKLGLAR (‘super-agonist’⁴), and a C5aR agonist, YSFKPMPLaR (‘EP54’⁷) were synthesized as previously described^{4,7}. It should be noted that the C5aR agonist also activates C3aR⁶ and thus is described herein as a dual C3aR/C5aR agonist. The agonists (1 $\mu\text{g}/\text{mg}$) were injected to the ventricles of the embryos together with the indicated plasmids.

Microscopy, quantification and statistical analyses

Images were taken using either Panoramic MIDI scanner (3DHistech) or confocal microscopy (LSM780, LSM800 Zeiss), equipped with Axio Observer Z1 microscope, and imaged with either Plan-apochromat 20x/0.8, or Plan-apochromat 40x/1.2, or Plan-apochromat 63x/1.4 oil objectives. The scaling data are 0.624X0.624 μm per pixel for 20X magnification, 0.312X0.312 μm per pixel for 40X magnification, and 0.198X0.198X0.51 μm per voxel for 60X magnification. The images were processed by ZEN software and/or Imaris software.

Cell count, positioning and colocalization analyses were performed using Imaris software (Bitplane Inc., Zurich, Switzerland, Imaris core module). For the cell count and positioning the relevant channel of an area of interest was analyzed with “Spots” module of Imaris, every spot labeling approximate center of the cell body. The “y” position of all the dots was analyzed by Microsoft Excel Histogram tool. Statistical analysis was performed by two-way analysis of variance (ANOVA) followed by Bonferroni multiple comparison analysis, using PRISM 7 for Mac (GraphPad software).

8. The data suggests that C1qa knockout doesn't cause defects in neuronal migration, however C1s knockdown does show a neuronal migration defect. C1s is part of a complex with C1qa, this discrepancy needs to be addressed.

We added to the discussion:

“Knockdown or knockout of specific elements of the classical pathway (C1qa and C1s) exhibited a mild effect on the progression of neuronal migration. The classical

pathway is initiated by the creation of the C1 complex which is composed of C1q, C1s and C1r. The formation of the complex is controlled by C1-inhibitor, which can physically interact with either C1s or C1r and thereby interfere with creation of the C1 complex. C1-inhibitor prevents the formation of the activation complexes of the lectin pathway and the thrombin pathway as well. *C1qa* KO mice do not have functional C1 complex, and do not exhibit a neuronal migration defect. Therefore, we concluded that the classical pathway is not involved in the migration regulation. Our findings implicated *C1s* in regulation of neuronal migration, it is possible that *C1s* has activities that are beyond those related to the classical pathway. For example, it may be possible to postulate that knockdown of *C1s* may lead to higher levels of unoccupied C1-inhibitor that may in turn inhibit the lectin pathway, and this over-inhibition results in neuronal migration impairment. Unlike the process of synapse elimination, where the classical pathway plays a major role^{8,9}, our results point to the greater importance of the lectin pathway in the regulation of neuronal migration.”

Minor concerns:

1. The authors use different developmental time points to demonstrate that C3, Masp1, and Masp2 are expressed during development in figure 1. This is curious and difficult to compare molecule expression patterns. Furthermore, this is not clear until one reads the legend carefully.

The immunostainings were unified to E16.

2. In Figures 2-4, it is unclear how cells are labeled for quantification. It is assumed that the authors are showing GFP labeled cells following electroporation and taking a % of GFP positive cells but this needs to be stated.

The labelling of “the % of GFP+ cells” is added to all relevant graphs.

3. In Figure 4 d and h, the authors do not include control shRNA controls on the graphs.

The controls were added to figure 4.

4. The authors show in situ hybridizations in supplemental figures 1 and 2. These data do not add much to the paper and are from an online database. They also do not have the proper KO controls, which are important for interpreting in situs.

The in situ hybridizations were removed from the figures.

5. Authors state that they used two-way ANOVAs throughout the study. There are clear cases where one-way ANOVAs are more appropriate (e.g., Figures 2&3).

One-way ANOVA is, most likely, not a more suitable statistical test, we have 3 characteristics: treatment, layer and the % of GFP+ cells, the significance of the tests will be almost identical when one-way ANOVA is conducted.

6. In Supplemental Figure 1, the authors include a western blot to show C3b. These data are weak and it is unclear why other C3 fragments are not also visualized. A clear positive control (such as serum) included on the blot would help.

A positive control (liver extract) is included.

7. Data referenced in that paper, such as ELISA and qPCR, are not shown. Please include these data.

Everything is written in the text:

ELISA

We next questioned whether the pathway is active in the developing brain. To this end, we tested and showed proteolytic processing of Complement protein C3 using an ELISA kit specific for the detection of cleaved C3 (C3a); we detected 2.89 ± 0.32 ng of cleaved C3 per mg of total protein with no C3a expression in C3 KO cortices.

qPCR

- The efficiency of C3 shRNA was confirmed by qPCR and was found to reduce the mRNA levels dramatically to $14.5 \pm 10.6\%$ ($n=6$; $p=0.009$) of the control expression levels.
- *Masp1* shRNA reduced *Masp1* mRNA levels by $38.5 \pm 3.5\%$ in comparison to control shRNA (real-time qPCR, $n=9$, $p=4.8E^{-08}$.)
- *Masp2* shRNA levels were reduced by $52.4 \pm 7.4\%$ in comparison to control ($n=9$, *Student's t-test* $p=7.1E^{-05}$).
- The reduction of *C1s* mRNA by shRNA expression was verified by qPCR, displaying $47.4 \pm 8\%$ reduction in the levels of *C1s* as compared to control shRNA ($n=5$, *Student's t-test*, $p=0.0067$).
- The C3a and the C5a receptors are expressed in the developing brain as evident by RNA *in situ* hybridization data of E14 brain from GenePaint.org and real-time RT PCR from E13-E18 cortices (supplementary Fig. 1l,m). The relative levels of C3aR and C5aR in the E16 cortex are $14.7 \pm 2.4\%$ and $2.9 \pm 0.4\%$, respectively, in comparison to the levels of C3aR and C5aR in adult mouse liver.

8. In the discussion, the authors claim, "The most persistent impairment of neuronal migration was observed in the case of *Masp1* shRNA, where at P8, treated neurons did not reach the cortical plate and acquired a fate of deep layers", yet they do not show P8 data for any other manipulation.

The data from all the shRNA manipulations is now added to figure 2 (Fig 2k-r) and also to supplementary figure 2 g-j.

In addition, pharmacological activation of the C3a and the C5a receptor in *Masp1* shRNA knockdown to ameliorate these effects, they only show data for C3 and *Masp2* knockdown.

The idea of the rescue experiments was to validate that the pathway is active, we believe that testing all genes with all possible rescue experiments is beyond the scope of

this study. Masp2 was chosen for rescue experiments because its functioning is restricted to the activation of the lectin pathway. In contrast, though Masp1 is known to be part of the lectin pathway, it was shown to activate the alternative pathway¹⁰ and even to cleave C3 by itself¹¹, thus presenting a more complex target when aiming to work with the pathway.

Added in the text:

Masp2 was chosen for rescue experiments because its functioning is more restricted to the activation of the lectin pathway in comparison with *Masp1*. *Masp1* has been shown to activate the alternative pathway¹⁰ and even to directly cleave C3¹¹.

9. In the discussion, the authors write "Knockout of lectin pathway components resulted in stronger migration phenotypes". Stronger than what? It is assumed they are referring to the classical pathway; however, there does seem to be a significant defect with C1s shRNA.

The discussion was modified.

10. The writing in the abstract, introduction, and discussion is often disjointed. The authors should revise to increase readability and fluidity.

The text has been extensively modified.

Reviewer #2 (Remarks to the Author):

The manuscript by Gorelik et al describes that some complement components are required for radial migration of cortical neurons. The authors first showed the expression pattern of the complement components, C3, MASP1 and MASP2 in developing mouse brains. Second, using in utero electroporation, the authors analyzed the consequences of either knockout or knockdown of key component of two branches of the complement cascade, the lectin and the classical pathway, on radial migration. They found that while the lectin pathway is likely regulating neuronal placement, the classical pathway is mildly involved. By analyzing the position of neuronal cell at later stages (P8), the authors showed that MASP1 depletion not only permanently impairs neuronal placement but also affects the specification of the misplaced neurons. Third, Gorelik and colleagues showed that both C3 cleavage and increased activation of the downstream receptors of the cascade (C5aR and C3aR) rescued the migration deficit observed in absence of MASP2 or C3. The authors finally identified de novo mutation in complement genes in ASD patients and, therefore, suggested an involvement of the complement pathway in ASD etiology.

Even though the authors do not propose any mechanism by which the lectin pathway could regulate neuronal migration, they provide strong evidences for a function of this complement pathway in the embryonic brain. However, the authors need to address a number of concerns to substantiate their conclusion.

Major points

1) As C3 and masp1 and masp2 seems to be expressed in progenitors as well, how the

authors are sure that the phenotype they observed is strictly due to migration defect and not to additional defect occurring in progenitors? Authors should moderate their conclusion and talk about positioning instead of migration; Induction of shRNA specifically in postmitotic neurons by in utero electroporation of CRE-inducible shRNA (for Masp1 or Masp2) should determine whether the abnormal migration results or not from defect in progenitors (see Garcez et al, nature communication, 2015).

According to the suggestions, we moderated our conclusions. We added additional experiments testing the possible effects on proliferation (Supplementary figure 3). Interestingly, knockdown of Masp1 did not show any effect in the proliferation assays we conducted. Knockdown of Masp2 and knockout of C3 had some effects on neuronal proliferation. These findings were supported by the reduced size of cortices in Masp2 and C3 knockout, but not in Masp1 knockout. We also added information regarding the morphology of the stalled cells, indicating that the cells are multipolar (Fig 2f-j). These results suggest that the cells are somewhat detained in the multipolar to bipolar transition.

2) Could the author discuss why the Migration/Position of Masp1 KD cells is only partially rescued by ectopic expression of Masp1? Same comment for figure 3b-g (C3 shRNA) and figure 3l-q (masp2 shRNA). This partial rescue could reflect off targeting of the shRNAs (of note, many shRNA (even ct shRNA) are shown to give off target effect by buffering endogenous miRNAs machinery (Baek et al, neuron, 2014)).

The possible off-targeting effect of shRNA constructs is a known problem, still the usage of shRNA remains one of the best instruments to manipulate specific genes at a specific time point. Controls are extremely most important in these kinds of experiments. The control shRNA uses the same machinery as the specific shRNA. Another control demonstrating the specificity of shRNA constructs is a rescue experiment. Achieving complete rescue is difficult, especially in view of the fact that many proteins require very precise amounts in migrating neurons. Indeed, dosage sensitivity has been demonstrated for several genes that are known to affect neuronal migration such as *Lis1*¹² and *Par1/Mark2*^{13, 14}.

In regard to our experiments, we tested the possible effects of overexpression of C3a and C3beta constructs. Both constructs interfered with neuronal migration (figure R2).

Figure R2: Brains were in utero electroporated (E14-E18) with GFP, C3a or C3beta constructs. There are more cells arrested in the IZ in case of C3a and C3beta in comparison to control. The scale is 50 μ m.

Does expression of Masp1 rescue the positioning of GFP cells in Masp1 KO mice?

As explained above, we did not establish a Masp1 KO line and cannot perform the suggested experiment.

On the same line, while the C3aR/C5aR agonist only partially rescues the phenotype upon C3 KD (fig 4e-h), it seems to completely rescue the phenotype in C3KO (fig 4i-k). (see also minor point 21)

Indeed, the double agonist managed to bring to complete rescue in C3 KO mice and only to the partial rescue in C3 KD, but the results can be interpreted otherwise: in C3 KO the injection of the double agonist resulted in additional 44% of cells that reached the upper layer compared to KO without the agonist. In the KD experiment additional 36% of cells managed to reach the upper layer following injection of the agonist. From this perspective, the effect of the agonist is similar in both systems.

3) The authors dismissed the role of the classical pathway in regulating neuronal positioning based on two results: 1/ no phenotype is observed in C1qa KO mice (fig 1 o-q); 2/ only a mild phenotype is observed upon C1s depletion (fig S1e-h). However:

- They haven't validated the complete depletion of C1qa in the C1qa KO brains.

The Cqa KO mice are complete KO, validated in many other studies (for example¹⁵⁻¹⁷).

- C1s shRNA reduces C1s by 48% in comparison with control shRNA. One cannot exclude to get a more striking phenotype using more efficient shRNAs. An alternative could be to use siRNAs.

The reduction of 48% was achieved in the mixed population of cells that only about 50-60% of them were transfected. Thus, the actual reduction in the transfected cells is higher.

In addition, the absence of a migration phenotype in C1qa KO mice suggests that even though C1s may influence neuronal displacement, it may be not through the standard activation of the classical pathway.

We added to the discussion:

"Knockdown or knockout of specific elements of the classical pathway (C1qa and C1s) exhibited a mild effect on the progression of neuronal migration. The classical pathway is initiated by the creation of the C1 complex which is composed of C1q, C1s and C1r. The formation of the complex is controlled by C1-inhibitor, which can physically interact with either C1s or C1r and thereby interfere with creation of the C1 complex. C1-inhibitor prevents the formation of the activation complexes of the lectin pathway and the thrombin pathway as well. C1qa KO mice do not have functional C1 complex, and do not exhibit a neuronal migration defect. Therefore, we concluded that the classical pathway

is not involved in the migration regulation. Our findings implicated C1s in regulation of neuronal migration, it is possible that C1s has activities that are beyond those related to the classical pathway. For example, it may be possible to postulate that knockdown of C1s may lead to higher levels of unoccupied C1-inhibitor that may in turn inhibit the lectin pathway, and this over-inhibition results in neuronal migration impairment. Unlike the process of synapse elimination, where the classical pathway plays a major role^{8,9}, our results point to the greater importance of the lectin pathway in the regulation of neuronal migration.”

4) Masp1 KD leads to permanent arrest of cells in deep layers. Could the authors comment on the following points:

- From pictures shown in figure 6i-j, it seems that ALL the Masp1 depleted cells are arrested in deep layers. Are those images representative? Would some cells still reach the CP or be dispersed throughout the CP? If yes, please provide more accurate pictures and quantification. If not, what are becoming the cells that migrate out the IZ at E18 (2c)? Do you observe any cell death? In the same line, if some cells reach the upper layers, are those cells well specified (cux1)?

The image was changed. There are the electroporated cells that reach the CP and are present there in P8. The anti-Cux1 immunostainings were added to the figure.

- Do the authors observed defect in layering in masp1 KO mice? Immunostaining of P2 or P8 Masp1 KO mice with layer specific markers should clarify this point.

The immunostainings of E18 slices of Masp1, Masp2 and C3 KOs were performed and were added to figure 1.

- Figure 2k-p: please provide pictures encompassing the entire cortical thickness.

The P8 immunostainings images were modified and now show the entire cortical width.

- immunostaining with upper layer markers (cux1, satb2) should be included to make sure that all the stalled cells are not correctly specified. If required, provide quantification.

Cux1 and Tbr1 immunostainings were performed to characterize the arrested cells in Masp1 shRNA and other shRNA treatments.

5) The authors showed a defect of specification. This defect can explain the impaired positioning of cortical neurons. One can imagine that the migration process by itself (multi-bipolar shape conversion, locomotion) is not affected. Could the author clarify what are the root causes of the phenotype?

The data showing impaired morphology of arrested neurons in all shRNA treatments were added to the figure 2 (Fig 2f-j). These data suggest that the multipolar-to-bipolar

transition may be affected following manipulation of different genes from the lectin pathway.

6) To analyse migration, the authors use in utero electroporation of GFP except in Masp1 and Masp2 KO mice where they use IdU birthdating (figure 1s-w). Could the authors explain why they are not using the same technique in every KO models? In utero electroporation of GFP plasmid in Masp1 and masp2 KO mice would greatly help to compare the different sets of experiments.

As explained above, the embryos were generated by injecting CRISPR/Cas9 at the one cell stage. We could not add to these mice an additional in utero electroporation experiment.

7) The authors showed that reduction by only 38 % of Masp1 induces a strong migration phenotype with very few depleted cells reaching the cortical plate (12%) compared to the control (figure 2a-d), while complete deletion leads to a milder phenotype (30% of cells in CP)(figure 1 u-w). Could the authors discuss this particular point?

We did not generate Masp2 and Masp1 KO lines, we analysed mutant embryos, each of which was a result of a single CRISPR/Cas9 injection in the fertilized oocyte. We agree that the KO and knockdown are not identical, however they do share common features. In most instances where KO and knockdown are studied in parallel, the knockdown phenotypes are more pronounced than the KO. This can be possibly explained by gene redundancy, which can occur in the KO but not the acute knockdown. A convincing and comprehensive study was performed on this topic¹.

Alternatively, or in addition, possible changes in the observed phenotype may be due to different techniques used in the analysis. The knockdown and knockout are analysed in two different and widely used methods. Whereas in case of KD in utero electroporation studies were performed in case of the KO cells were labelled by IdU. GFP in utero electroporation usually results in a more uniform label of cells, whereas in case of IdU it is possible that we are analysing also cells that underwent one cell division. It is difficult to distinguish between the first and second generation of cells and this results in a more dispersed presentation of the cells. Even though the techniques differ, we do not claim that the phenotypes are the same.

Historically, we initiated our studies using knock-down systems because:

- a) the experiments with ICR mice require less animals to be sacrificed for each experiment with adequate results compared to C57/Bl6 mice.
- b) Using knock-down we can introduce several experimental combinations (knock-down, cell labelling, rescue constructs).
- c) Especially when we are analysing a pathway, which involves multiple genes, knock-down systems are much more time-efficient versus the time required to

generate multiple knock-out genes (especially when we are discussing the pre-CRISPR/Cas9 era.

d) The only mouse line that was available for our studies was C3 knockout.

e) We generated Masp1 and Masp2 knock-out using CRISPR/Cas9 gene editing technology and analysed embryos directly. This required injecting the pregnant mice with BrdU or IdU and using all the embryos generated by this technique, genotyping, and immunostaining. Each embryo is an independent targeted event and several embryos were analysed in each case. We did not keep the lines.

8) Masp2, the closest paralog of Masp1 partially rescued the phenotype in Masp1 KD neurons (Fig 2g-h), suggesting functional overlap between those two proteins. One protein could compensate for the loss of the other in vivo. Could this explain the mild phenotype in masp1 or masp2 KO mice compared to C3 KO mice? To test this hypothesis, authors could perform GFP in utero electroporation in masp1/masp2 double KO and test if they worsen the phenotype compared to single KO. Are masp1 or masp2 more expressed in masp2 or masp1 KO brains, respectively?

This is very plausible explanation. Unfortunately, as explained earlier, no mice lines are available to perform the suggested experiments.

9) In Figure 1, the authors analyzed the expression of C3, Masp1 and Masp2 in the developing cortex. While C3 expression/localization has been assessed at E16.5, the masp1 and masp2 immunostainings have been performed at E14.5 and E18.5. Could the authors explain their choices?

The immunostainings were unified to E16.

Does that mean that C3 is not or slightly expressed at later time points? It would have been more informative to: 1) analyze by western blot (or alternatively qPCR) the amount of each protein at those different developmental stages to test whether their expression level change over the time;

The qRT-PCR data were added to supplementary figure 1, showing the presence of mRNA of all discussed genes throughout the E13-E18 time window with some fluctuations in between the different time points.

and 2) show pictures of entire thickness of the cortex at each stage instead of magnification at E18.5. On the same line, one doesn't know which part of the cortex (CP, VZ, IZ) is shown in 1e and 1h.

The figure was changed to show the entire width of the cortex (Supplementary figure 1).

10) Figure S1A shows a decreased cleavage of C3 as the development progresses. Could the authors comment about a possible reduction of complement activity throughout the development? Could this decrease result from a reduction in C3 protein level?

The gel presented previously in the supplementary figure 1 did not include the loading control, so it was impossible to conclude about the changes in the protein amount. The original purpose of the western blot was to demonstrate that the cleavage occurs in the cortex. The new western blot includes a loading control, and shows slight fluctuations of C3b levels among developmental days.

11) As C3 cleavage products only partially rescue masp2 KD phenotype, could the authors discuss other possible function of masp2?

As shown above and below (Fig R2), overexpression of C3 cleavage products may inhibit neuronal migration, it is possible that the expression of these proteins is not optimal.

Figure R2: Brains were in utero electroporated (E14-E18) with GFP, C3a or C3beta constructs. There are more cells arrested in the IZ in case of C3a and C3beta in comparison to control. The scale is 50 μ m

12) As the entire rescue experiments are performed in masp2 shRNA electroporated cells, why do the authors analyze postnatal consequences after masp1 KD in figure 2? It would have been better to perform those analyses in the same model.

The rescue experiments were done in C3 KD and C3 KO as well as in Masp2 shRNA treated brains.

We preferred Masp2 over Masp1 for the reasons described above (answer to reviewer 1 minor 8)

The postnatal data for Masp1, Masp2 and C3 are included in the new version of the figure 2.

Minor points

13) While the text indicates that the immunostainings of C3 have been performed at both E14,5 and E18,5, the figure legend mentions E16,5.

The figure and the text were changed.

14) Figure 1 legend: could the authors homogenize the developing stages (E14,5 or E14, E18,5 or E18)

The figure and the text were changed.

15) Figure 1e-g: the authors claim that "high levels (of masp1) were noted in the VZ, upper IZ and upper CP. Colocalisation with specific markers for those region will strengthen this conclusion.

The markers were added to the figure.

16) Masp1 and Masp2 KO mice: although those models have been used only in figure 1, a full validation of those models engineered using the CRIPR/Cas9 technology is missing. Has the authors assessed possibility of off targets?

Each of the embryos were validated for the lack of the respective protein expression. Three embryos or more were used for analysis, each embryo is derived from an independent CRISPR event, therefore, it is unlikely that the same off targets will be affected.

17) Supplemental document: IdU injections section: BrDU is mentioned??

Changed to IdU, however the anti-BrdU antibodies are cross reactive with IdU and this is their commercial name.

18) For greater clarity, S1f and h should be merged (as presented in figure 4). Same comment for figure 2b-d-f -h; 3-c-e-g-l-k and 3m-o-q-s-u. It will help comparisons.

The graphs were merged

19) The authors show statistics in comparison to shRNA condition. Adding statistics in comparison to ct condition would help to decipher between total or partial rescue. (figure 2b-d-f -h; 3-c-e-g-l-k and 3m-o-q-s-u)

The statistical analysis was changed and added to the graphs.

20) As control, the authors performed GFP or Ctl shRNA in utero electroporation in WT mice. Could the authors comment the difference of migration profile in those two types of control (70 % and 50% of cells reaching the CP after GFP or shRNA electroporation, respectively?

Could the authors clarify in which mouse background in utero electroporations of shRNA have been performed?

All shRNA experiments were performed on ICR mice. Mentioned in the methods.

21) For greater clarity, authors should include ct condition in their graph (figure 4).

Included

Reviewer #3 (Remarks to the Author):

In this manuscript, Gorelik et al clearly demonstrate that activation of the lectin arm of the complement pathway regulates neuronal migration. These studies not only provide new insights into the mechanisms of neuronal migration, but also underscore the relevance of this cortical developmental process to ASD. It shows that immune system malfunctions during development may exert much wider effect on brain development than previously thought. The authors have used comprehensive developmental expression, gene knockdown/ knock out studies of complement pathway genes (C3, Masp1, 2) to illustrate this. In general, this study is logically organized, clearly illustrated, and provides much needed new information on immune system-brain development cross talk. Addressing the following points will be necessary to strengthen the clarity and impact of this work.

1) While its clear that C3/ Masp 1, 2 knockout or knock down affects neuronal migration, some of the reduced neuronal numbers/ placement effects may have ben due to altered proliferation of cortical progenitors in these experimental models. Please provide evidence to rule out this possibility.

We performed the experiments to address the possible effects on proliferation. The data are present in the supplementary figure 3.

2) In C3 and Masp1, 2 null mice, please show altered migration/ placement with standard cortical layer specific neuronal markers (partially addressed in Fig.2, but not adequately).

The required data are added to figure 1 (Fig 1u-af).

3) Does activation of the lectin arm of complement pathway cause death of migrating neurons?

To check whether manipulations with the lectin pathway influence apoptosis we performed immunostainings with anti-cleaved Caspase 3 Abs on slices from in utero electroporated brains (E14-E18) with control or Masp2 shRNA. A figure for the reviewer is added below (figure R3). No difference was observed.

Figure R3: Brains were *in utero* electroporated (E14-E18) with control or Masp2 shRNA. The immunostainings with anti-cleaved Caspase3 antibodies were performed. The scale bar is 50 μ m.

4) While its admittedly beyond the scope of this work, do the authors have any evidence to show that the damaging mutations detected in C3 (K610T and D1569E) and in MASP1 (Y294N) affects neuronal migration. Such information will provide direct evidence for the connection between complement pathway, neuronal migration, and ASD, currently missing from this work.

We realize that and have moved the information about the ASD related *de novo* mutations to the discussion.

1. Rossi, A. *et al.* Genetic compensation induced by deleterious mutations but not gene knockdowns. *Nature* (2015).
2. Ember, J.A., Johansen, N.L. & Hugli, T.E. Designing synthetic superagonists of C3a anaphylatoxin. *Biochemistry* **30**, 3603-3612 (1991).
3. Proctor, L.M. *et al.* Complement factors C3a and C5a have distinct hemodynamic effects in the rat. *Int Immunopharmacol* **9**, 800-806 (2009).
4. Wu, M.C. *et al.* The receptor for complement component C3a mediates protection from intestinal ischemia-reperfusion injuries by inhibiting neutrophil mobilization. *Proc Natl Acad Sci U S A* **110**, 9439-9444 (2013).
5. Kawatsu, R. *et al.* Conformationally biased analogs of human C5a mediate changes in vascular permeability. *J Pharmacol Exp Ther* **278**, 432-440 (1996).
6. Scully, C.C. *et al.* Selective hexapeptide agonists and antagonists for human complement C3a receptor. *Journal of medicinal chemistry* **53**, 4938-4948 (2010).
7. Woodruff, T.M. *et al.* Species dependence for binding of small molecule agonist and antagonists to the C5a receptor on polymorphonuclear leukocytes. *Inflammation* **25**, 171-177 (2001).

8. Stevens, B. *et al.* The classical complement cascade mediates CNS synapse elimination. *Cell* **131**, 1164-1178 (2007).
9. Schafer, D.P. *et al.* Microglia sculpt postnatal neural circuits in an activity and complement-dependent manner. *Neuron* **74**, 691-705 (2012).
10. Selander, B. *et al.* Mannan-binding lectin activates C3 and the alternative complement pathway without involvement of C2. *J Clin Invest* **116**, 1425-1434 (2006).
11. Matsushita, M. & Fujita, T. Cleavage of the third component of complement (C3) by mannose-binding protein-associated serine protease (MASP) with subsequent complement activation. *Immunobiology* **194**, 443-448 (1995).
12. Bi, W. *et al.* Increased LIS1 expression affects human and mouse brain development. *Nat Genet* **41**, 168-177 (2009).
13. Sapir, T. *et al.* Accurate balance of the polarity kinase MARK2/Par-1 is required for proper cortical neuronal migration. *J Neurosci* **28**, 5710-5720 (2008).
14. Sapir, T. *et al.* Antagonistic effects of doublecortin and MARK2/Par-1 in the developing cerebral cortex. *J Neurosci* **28**, 13008-13013 (2008).
15. Botto, M. C1q knock-out mice for the study of complement deficiency in autoimmune disease. *Exp Clin Immunogenet* **15**, 231-234 (1998).
16. Bulla, R. *et al.* C1q acts in the tumour microenvironment as a cancer-promoting factor independently of complement activation. *Nat Commun* **7**, 10346 (2016).
17. Naito, Atsuhiko T. *et al.* Complement C1q Activates Canonical Wnt Signaling and Promotes Aging-Related Phenotypes. *Cell* **149**, 1298-1313 (2012).

Reviewers' comments:

Reviewer #2 (Remarks to the Author):

The manuscript by Gorelik et al, entitled "Unexpected Activities of the Complement Pathway in Migrating Neurons" provides evidence that proteins in the lectin arm of the complement pathway, specifically MASP1, MASP2, and C3, are expressed in the brain during embryonic development and developing mice deficient in these molecules have disruptions in neuronal migration and proliferation. Further, molecular mimics of C3 cleavage products and agonists for C3aR and C5aR rescue defects. The manuscript offers an exciting and novel mechanism by which complement regulates nervous system development with implications for ASDs. The authors have made significant strides towards addressing reviewer concerns. However, there are still remaining issues outlined below:

1.) The explanation for the large variability between Masp1 and Masp2 knockout (Fig 1) versus shRNA knockdown (remaining figures) is speculative and not convincing. The authors raise the point that the CRISPR/cas9 strategy may result in genetic redundancy. Alternatively, it could introduce off-target effects. With such ambiguity, these data are better suited to supplement or removed. As it is now, these data confuse the more clear shRNA results.

2.) The validation of shRNA knockdown and genetic rescue still require more work. The authors provide un-quantified data in cultured neurospheres which was in the original manuscript (supplemental figure 2) and provide qPCR results in the text. If possible, validation should be performed in vivo, quantified, and assessed at the level of protein. The authors state that they do not have sufficient antibodies to perform validation experiments. While this is understood to be a sufficient road block, it is unclear why they could not use their western blotting and immunofluorescence techniques used in supplementary figure 1 to validate shRNA knockdown.

3.) The authors have not provided evidence that electroporation efficiency is similar across experiments. The authors stated in their response, "We exclude brains in which only a small number of cells are GFP positive". How do the authors determine what is a small number of cells? It is understood that results are presented as a % of GFP cells, which should then provide some normalization for variability across animals. However, if there are cell-extrinsic effects (as suggested by extracellular localization of C3), the number of cells transduced could have a large impact on cell migration/proliferation.

4.) Given the new cell proliferation data in supplemental Fig 3, the authors should more clearly outline how migration and, albeit to a lesser extent, proliferation are both affected. They do state, "The observed changes in proliferation of the progenitors may explain in part the reduction in the width of C3 and Masp2 knockout cortices". The authors did not explain clearly how increased proliferation could lead to a decrease in cortex width? The authors should consider moving these proliferation data to a main figure, further discuss these data, and include data for all the animals in the graphs.

5.) The postnatal data in Fig 2k-r is a nice addition. However, the authors need to provide quantification.

6.) There is a discrepancy in one data set. In figure 1n, the percent of GFP+ cells in the cortical plate of C3 KO is about 40%. In figure 4n, the percent of GFP+ cells in the cortical plate is about 55%. What accounts for the 15% difference?

7.) Experimental methods still need more detail. For example, the authors do not explain how they pick regions for quantification of immunofluorescence images, how many regions they sample per animal etc. Western blot methodology is also missing and it is unclear whether the authors perfuse animals with PBS to remove blood prior to performing ELISA or western blot. This is particularly important given that these proteins are enriched in serum.

8.) The authors provide convincing figures in their response, figure R1 and R3. These data should be quantified and added to the manuscript as supplementary figures.

9.) The authors state in their response that one-way ANOVAs are not appropriate. In the original version, many graphs appeared to compare % of GFP cells across regions versus also comparing

conditions (KO vs WT, etc.). This was revised in the current manuscript and conditions were added to the graphs. It is agreed that two-way ANOVAs are now most appropriate. However, there are still some graphs in which one-way ANOVAs should be used given they are comparing one condition (e.g. KO vs WT) without any other parameter (e.g. CP, VZ, etc.): Fig 1w,x,aa,ab,ae,af; Fig 2I, supplementary Fig 1i-m, supplementary Fig 3d.

10.) The authors should devote a larger part of the discussion to what could be downstream of complement to regulate cell migration/proliferation. There is one line describing the possibility that cytoskeleton could be modified but no references or further discussion of known cytoskeletal pathways regulated by complement, etc. Also, please add more discussion regarding how neuronal migration/proliferation and ASDs could be linked

11.) While it is appreciated that titles should be succinct, the current title is somewhat ambiguous. It should be revised to more clearly describe the findings.

Reviewer #3 (Remarks to the Author):

The revised manuscript has been largely improved. Most of my concerns have been addressed. As a minor point I would like to point out that, in the paper describing the C1q KO mouse model (botto, nat genet, 1998, ref 15 of the current ms), the complete depletion in the brain has not been assessed (neither in the papers mentioned in the rebuttal letter). We cannot rule out the possibility of some hypomorphism in the brain (differential splicing). More importantly, some experiment that have been added in this revised manuscript raise a new concern: to address the specification of the stalled neurons, the authors have performed co labeling of GFP with either Cux1 or Tbr1. They conclude that :

- "the stalled C3, masp2 and masp2 shRNA treated cells were Cux1 positive": from the images it seems that some of the arrested masp2-silenced cells are not expressing cux1 (red inset in figure 2n).
- "both control and shRNA treated cells were negative for the deep layer neuronal marker Tbr1": again from the images (no quantification provided) it seems that most of the masp1 depleted cells and some of the masp2 depleted cells express Tbr1 when arrested in the IZ. The authors should comment on those new results that suggest a defect in specification!

Reviewer #4 (Remarks to the Author):

The authors have addressed all of this reviewer's concerns with new data. Nice job. This manuscript is ready and appropriate for publication in Nature Communications.

We thank the reviewers for taking the time and effort to improve our manuscript. We hope that our revision properly addresses the comments raised by the reviewers. Please find our reply in a detailed point-to-point response.

Reviewer #2 (Remarks to the Author):

The manuscript by Gorelik et al, entitled "Unexpected Activities of the Complement Pathway in Migrating Neurons" provides evidence that proteins in the lectin arm of the complement pathway, specifically MASP1, MASP2, and C3, are expressed in the brain during embryonic development and developing mice deficient in these molecules have disruptions in neuronal migration and proliferation. Further, molecular mimics of C3 cleavage products and agonists for C3aR and C5aR rescue defects. The manuscript offers an exciting and novel mechanism by which complement regulates nervous system development with implications for ASDs. The authors have made significant strides towards addressing reviewer concerns. However, there are still remaining issues outlined below:

- 1.) The explanation for the large variability between Masp1 and Masp2 knockout (Fig 1) versus shRNA knockdown (remaining figures) is speculative and not convincing. The authors raise the point that the CRISPR/cas9 strategy may result in genetic redundancy. Alternatively, it could introduce off-target effects. With such ambiguity, these data are better suited to supplement or removed. As it is now, these data confuse the more clear shRNA results.

The partial discrepancies between knockout and knockdown models are known and addressed differently in different studies (Yang X., PNAS, 2013 vol 110, <http://www.pnas.org/content/110/51/20777.full>; A very convincing study (Rossi et al, 2015) in zebrafish model proves that at least in case of *egfl7* there is a gene redundancy effect in knockout versus knockdown. With all existing knowledge of the absence of complete similarity between knockout and knockdown, both techniques are used in research in many cases interchangeably (<http://www.nature.com/nature/journal/v514/n7521/pdf/nature13683.pdf>). In our study we did not observe contradictions between two utilized methods, the only difference is a magnitude of the observed effects. This is the reason why we consider that the usage of two different techniques to manipulate levels of C3, Masp1 and Masp2 is an advantage of our study. If one can claim for a possible off-target effect in one of the techniques, the chance for similar off-target using a completely different methodology is negligible.

The independent production of all CRISPR/cas9 embryos used in the study strongly suggests that the effect of off-targets is minimal.

- 2.) The validation of shRNA knockdown and genetic rescue still require more work. The authors provide un-quantified data in cultured neurospheres which was in the original manuscript (supplemental figure 2) and provide qPCR results in the text. If possible, validation should be performed in vivo, quantified, and assessed at the level of protein. The authors state that they do not have sufficient antibodies to perform validation experiments. While this is understood to be a sufficient road block, it is unclear why they could not use their western blotting and immunofluorescence techniques used in supplementary figure 1 to validate shRNA knockdown.

A new experiment was conducted to address this specific issue, please see Supplementary figure 2. *In utero* electroporation (E14-E16) of control shRNA, C3 shRNA, C3 shRNA+C3-resistant, Masp1 shRNA, Masp1 shRNA+Masp1 resistant, Masp2 shRNA and Masp2 shRNA+Masp2-resistant was performed. The brain slices were immunostained with anti-C3, anti-MASP1, and anti-MASP2 antibodies. The fluorescence signal of individual cells (outlined with GFP-LiveAct) was measured with ImageJ and compared. Our results demonstrate a reduction in the level of the immunostained proteins following shRNA treatment and partial (C3 and Masp1) or complete (Masp2) rescue of the levels of the indicated protein when shRNA resistant constructs were expressed.

- 3.) The authors have not provided evidence that electroporation efficiency is similar across

experiments. The authors stated in their response, "We exclude brains in which only a small number of cells are GFP positive". How do the authors determine what is a small number of cells? It is understood that results are presented as a % of GFP cells, which should then provide some normalization for variability across animals. However, if there are cell-extrinsic effects (as suggested by extracellular localization of C3), the number of cells transduced could have a large impact on cell migration/proliferation.

In our study we referred to a brain as a positive one, when the GFP signal can be easily seen in the embryo using a stereomicroscope prior to perfusion. We agree that taking into account the non cell-autonomous function of the studied proteins, the amount/density of electroporated cells is important. However, this density cannot be extracted directly from the number of electroporated cells per slice. The cortex has a 3D structure and cells born in VZ in a specific area migrate to the cortical plate and occupy a larger area. When we analyse slices, we do it in a rectangle form of a constant width ignoring larger dispersion of cells in upper layers compared to lower layers. This way of analyses is used in many studies. Obviously, when cells are retained in the VZ/IZ zone, less cells will reach the CP in comparison to treatments when most or all cells reach the CP, even if the total number of electroporated cells is identical.

4.) Given the new cell proliferation data in supplemental Fig 3, the authors should more clearly outline how migration and, albeit to a lesser extent, proliferation are both affected. They do state, "The observed changes in proliferation of the progenitors may explain in part the reduction in the width of C3 and Masp2 knockout cortices". The authors did not explain clearly how increased proliferation could lead to a decrease in cortex width? The authors should consider moving these proliferation data to a main figure, further discuss these data, and include data for all the animals in the graphs.

Our proliferation studies are limited and are not sufficient to reach a complete understanding of the role of the complement pathway in this process. To gain detailed understanding, more experiments, time-points and rescue experiments will be needed. In addition, further analysis of the affected proliferation events should be done. From our data meanwhile we can conclude that in Masp2 knockdown and in C3 knockout there observed changes in proliferation. We could not conclude whether the proliferation is increased. The increase of IdU+GFP+ cells in Masp2 knockdown means that there is an increase in the number of cells undergoing S-phase. It is possible that there are more cells entering a normal proliferation cycle or, alternatively, that some cells spend more time at the DNA synthesis stage. If the second option is correct, it is likely that the total number of cells will be reduced, subsequently leading to a thinner cortex. These same considerations are applicable to the observed increased number of pHis+ cells in C3 KO.

5.) The postnatal data in Fig 2k-r is a nice addition. However, the authors need to provide quantification.

The quantification of electroporated cells positions was added to Figure 2 (n). The quantification of immunostained-positive cells was added to Supplementary figure 2 (v-x).

6.) There is a discrepancy in one data set. In figure 1n, the percent of GFP+ cells in the cortical plate of C3 KOs is about 40%. In figure 4n, the percent of GFP+ cells in the cortical plate is about 55%. What accounts for the 15% difference?

These are two independent experiments. In both cases the control was conducted in parallel with the experiment. One of the technical discrepancies between these two experiments was that for the first one mating of C3KOXC3KO and C3WTXC3WT were performed, whereas for the second experiment C3HETXHET mating was used. We tested the effect of maternal genotype on embryo phenotype and we could not determine any correlation. Still it is possible that genotype of surrounding embryos may slightly modify the migration phenotype.

7.) Experimental methods still need more detail. For example, the authors do not explain how they pick regions for quantification of immunofluorescence images, how many regions they sample per animal etc. Western blot methodology is also missing and it is unclear whether the authors perfuse animals with PBS to remove blood prior to performing ELISA or western blot. This is particularly important given that these proteins are enriched in serum.

We added the required details to the methods. We did not perfuse animals for the mentioned experiments. In both cases, we removed the meninges and the choroid plexus that is the main source of blood in the developing cortex and surgically isolated the cortices. It should be noted that the amount of blood in isolated cortices during embryogenesis is minimal.

8.) The authors provide convincing figures in their response, figure R1 and R3. These data should be quantified and added to the manuscript as supplementary figures.

The data were quantified and Supplementary figure 3 and 7 were added.

9.) The authors state in their response that one-way ANOVAs are not appropriate. In the original version, many graphs appeared to compare % of GFP cells across regions versus also comparing conditions (KO vs WT, etc.). This was revised in the current manuscript and conditions were added to the graphs. It is agreed that two-way ANOVAs are now most appropriate. However, there are still some graphs in which one-way ANOVAs should be used given they are comparing one condition (e.g. KO vs WT) without any other parameter (e.g. CP, VZ, etc.): Fig 1w,x,aa,ab,ae,af; Fig 2l, supplementary Fig 1i-m, supplementary Fig 3d.

Fig 1w,x,aa,ab,ae,af – Student t-test was performed since this is a comparison between two groups, the information is added to text and to the legends.

Fig2l (Suppl Fig. 2q) – One-way ANOVA was performed. Added to legend

Suppl Fig 1 i-m – One-way ANOVA was performed. Added to legend

Suppl 3d (Suppl 5d) - One-way ANOVA was performed. Added to legend

10.) The authors should devote a larger part of the discussion to what could be downstream of complement to regulate cell migration/proliferation. There is one line describing the possibility that cytoskeleton could be modified but no references or further discussion of known cytoskeletal pathways regulated by complement, etc. Also, please add more discussion regarding how neuronal migration/proliferation and ASDs could be linked

Please note the following sections:

“Activation of the C3a or the C5a receptors are likely to activate small GTPases, which will in turn affect cytoskeletal dynamics required for proper neuronal migration. As key regulators of actin and microtubule cytoskeletons, cell polarity, and adhesion, the Rho GTPases play critical roles in CNS neuronal migration (reviews^{26,27}). The complement fragment C3a and its receptor have been previously demonstrated to act during collective cell migration of neural crest cells⁸. The mode of collective migration is rather different from radial neuronal migration studied here, although our studies imply that at least one of the chemo-attractants may be shared in both processes. In addition, perturbations in C5aR signaling during rodent brain development have been reported to result in select

defects^{28,29}, suggesting widespread roles for complement fragments C3a and C5a in neuronal development.”

“Consistent with this idea, database searches indicated that multiple complement pathway components are associated with ASD with different types of associations³⁷. These disorders are typically characterized by social deficits, communication difficulties, stereotyped or repetitive behaviors and interests, and in some cases, cognitive delays. Neuronal migration deficits should be considered as one of the underlying pathologies in ASD¹¹. New exome sequencing technology has identified novel rare variants and has found that sporadic cases of ASD are enriched for disruptive *de novo* mutations³⁸. Therefore, we further analyzed existing ASD exome sequencing data and show that ASD patients exhibit *de novo* mutations in complement genes³⁹⁻⁴¹ (Table 1). Mutations in 14 different genes were detected, additional evidence for the involvement of three of the genes in ASD exists (*COLEC12*, *C3* and *FGA*). Among the 16 identified mutations, 15 were analyzed by functional prediction programs, 6 or 9 were found to be damaging or possibly damaging by SIFT or PolyPhen program, respectively. Nevertheless, it is likely that not only mutations that are predicted to be damaging can be involved in disease, as has been recently demonstrated in patients with ASD or schizophrenia⁴². Underscoring our functional studies are damaging mutations detected in *C3* (K610T and D1569E) and in *MASPI* (Y294N). Collectively, our results and the abovementioned *de novo* mutations studies warrant further investigation to probe the involvement of the complement pathway in ASD.”

11.) While it is appreciated that titles should be succinct, the current title is somewhat ambiguous. It should be revised to more clearly describe the findings.

Changed to “Developmental Activities of the Complement Pathway in Migrating Neurons”

Reviewer #3 (Remarks to the Author):

The revised manuscript has been largely improved. Most of my concerns have been addressed. As a minor point I would like to point out that, in the paper describing the C1q KO mouse model (botto, nat genet, 1998, ref 15 of the current ms), the complete depletion in the brain has not been assessed (neither in the papers mentioned in the rebuttal letter). We cannot rule out the possibility of some hypomorphism in the brain (differential splicing).

Unfortunately, we don't have mice any more to address this question in an accurate way. We used perfused fixed with PFA E18 brains of C1qa KO and relevant WT control. We prepared RNA according to the standard protocol and performed Real-time PCR using C1qa primers published in Lobsiger et al., 2013; PNAS 110(46): E4385-E4392. The data was normalized to 29rps. Real-time data showed significant reduction of C1qa mRNA level in C1qa KO brains compared to WT ($2.65 \pm 0.73\%$, $n=2$, $p=0.002$), suggesting that at least at E18 there is C1qa depletion from the brain tissue.

More importantly, some experiment that have been added in this revised manuscript raise a new concern: to address the specification of the stalled neurons, the authors have performed co labeling of GFP with either Cux1 or Tbr1. They conclude that :

-“ the stalled C3, masp2 and masp2 shRNA treated cells were Cux1 positive”: from the images it seems that some of the arrested masp2-silenced cells are not expressing cux1 (red inset in figure 2n).

- "both control and shRNA treated cells were negative for the deep layer neuronal marker Tbr1": again from the images (no quantification provided) it seems that most of the masp1 depleted cells and some of the masp2 depleted cells express Tbr1 when arrested in the IZ. The authors should comment on those new results that suggest a defect in specification!

Quantification was performed and added to the supplementary figure 2. Indeed, there was a reduction in total proportion of Cux1+ cells in Masp1 and Masp2 knockdown. These Cux1- cells did not adopt a Tbr1+ or GFAP+ cell fate.

Tbr1 immunostainings have high background. Only nuclei that showed complete, not-dotted staining were considered positive for Tbr1.

Reviewer #4 (Remarks to the Author):

The authors have addressed all of this reviewer's concerns with new data. Nice job. This manuscript is ready and appropriate for publication in Nature Communications.

REVIEWERS' COMMENTS:

Reviewer #2 (Remarks to the Author):

The authors have sufficiently addressed all concerns.

Reviewer #3 (Remarks to the Author):

this last version of the manuscript is addressing reviewer concerns and is suitable for publication in *nat* communications.